# ISOM 1.0: A fully mesoscale-resolving idealized Southern Ocean model and the diversity of multiscale eddy interactions

Jingwei Xie[1], Xi Wang[2], Hailong Liu[1,3], Pengfei Lin[3,5], Jiangfeng Yu[3,5], Zipeng Yu[3], Junlin Wei[4,5,6], and Xiang Han[4,5]

[1]Laoshan Laboratory, Qingdao, China
[2]Aerospace Newsky Technology Co. LTD., Beijing, China
[3]Institute of Atmospheric Physics, Chinese Academy of Sciences, Beijing, China
[4]Computer Network Information Center, Chinese Academy of Sciences, Beijing, China
[5]University of Chinese Academy of Sciences, Beijing, China
[6]Pengcheng Laboratory, Shenzhen, China

**Correspondence:** Jingwei Xie (xiejw23@mail3.sysu.edu.cn) and Hailong Liu (hlliu2@qnlm.ac)

**Abstract.** We describe an idealized Southern Ocean model (ISOM 1.0) that contains simplified iconic topographic features in the Southern Ocean and conduct a fully mesoscale-resolving simulation with the horizontal resolution of 2 km based on the Massachusetts Institute of Technology general circulation model. The model obtains a fully developed and vigorous mesoscale eddying field with a $k^{-3}$ eddy kinetic energy spectrum and captures the topographic effect on stratification and large-scale flow. To make a natural introduction of large eddy simulation (LES) methods into ocean mesoscale parameterization, we propose the concept of mesoscale ocean direct numerical simulation (MODNS). A qualified MODNS dataset should resolve the first baroclinic deformation radius and ensure that the affected scales by the dissipation schemes are sufficiently smaller than the radius. Such datasets can serve as the benchmark for *a priori* and *a posteriori* tests of LES schemes or mesoscale ocean large eddy simulation (MOLES) methods into ocean general circulation models. The 2-km simulation can meet the requirement of MODNS and also capture submesoscale effects. Therefore, its output can be a type of MODNS and provide reliable data support for relevant *a priori* and *a posteriori* tests. We demonstrate the diversity of multiscale eddy interactions, validate the crucial role of mesoscale-related strain in submesoscale processes, and uncover the bridge effect of submesoscale processes between mesoscale entities and in the eddy-jet interaction. In addition, we use the model to conduct multipassive tracer experiments and reveal guidelines for the initial settings of passive tracers to delay the homogenization process and ensure the mutual independence of tracers over a long period.

## 1 Introduction

Oceanic mesoscale processes have motions with spatial scales of O(10km)-O(100km) or near the baroclinic Rossby deformation radius, including quasi-geostrophic eddies and meandering jets (Chelton et al., 1998; Hallberg, 2013; Siedler et al.,

2013; Thompson and Naveira-Garabato, 2014; Youngs et al., 2017). These processes encompass more than 80% of the oceanic kinetic energy and play a crucial role in material transport, heat transport, momentum budget, and air-sea interactions. They also modulate climate variability across multiple timescales and engage in drastic multiscale interactions with both large-scale and submesoscale processes (Stammer, 1998; Ferrari and Wunsch, 2009; Zhai et al., 2010; Chelton et al., 2011; Dong et al., 2014; Ma et al., 2016; Vallis, 2017; Busecke and Abernathey, 2019; Schubert et al., 2020; Taylor and Thompson, 2023; Yang et al., 2024). Fully resolving oceanic processes at the mesoscale requires ocean general circulation models (OGCMs) with kilometer-scale horizontal resolution (Marques et al., 2022). Such models require massive computational and storage resources for long-term integration or large-ensemble experiments. Therefore, we still need parameterizations that capture the collective effects of the unresolved parts related to oceanic mesoscale processes in lower-resolution OGCMs.

The classic works on parameterizing oceanic mesoscale processes include the isoneutral diffusion scheme by Redi (1982) and the revolutionary Gent-McWilliams (GM) scheme (Gent and McWilliams, 1990; Gent et al., 1995) that represents the effect of eddy-induced adiabatic advection as well as generating a net sink of available potential energy. These two schemes are widely used in coarse-resolution OGCMs and can be represented by a flux-gradient relationship with an asymmetric transport tensor (Griffies et al., 1998). Since its inception, scholars have made considerable advancements in the field based on the GM-Redi framework. Additional specific constraints or properties, such as the stratification state (Visbeck et al., 1997), anisotropy (Smith and Gent, 2004), geometrical information (Mak et al., 2018; Wei et al., 2024), and energetic constraints (Cessi, 2007; Eden and Greatbatch, 2008; Jansen and Held, 2014; Mak et al., 2018; Bachman, 2019; Jansen et al., 2019; Yankovsky et al., 2024), have been embedded in the scheme to produce spatiotemporal variations in the transport coefficients.

In addition to the traditional theory-driven schemes described above, other studies have revealed the potential of large eddy simulation (LES) methods in oceanic mesoscale parameterization (Fox-Kemper and Menemenlis, 2008; Graham and Ringler, 2013; Pearson et al., 2017; Khani et al., 2019; Khani and Dawson, 2023; Xie et al., 2023; Perezhogin and Glazunov, 2023). The application is sometimes called the mesoscale ocean large eddy simulation (MOLES) (Fox-Kemper and Menemenlis, 2008; Graham and Ringler, 2013). First, one advantage of LES is in addressing the Reynolds averaging issue (Khani and Dawson, 2023; Xie et al., 2023; Perezhogin and Glazunov, 2023). Many parameterizations and related diagnostics originating from the GM-Redi framework are based on Reynolds averaging, which may simplify the derivation. However, the Reynolds averaging method inherently suppresses cross-scale interactions near the grid scale, leading to a loss of local information, and its mathematical properties are not fully satisfied by the grid discretization of numerical models (Leonard, 1974; Germano et al., 1991; Germano, 1992; Pope, 2000; Xie et al., 2023). Using Reynolds averaging as a grid discretization approximation in very coarse-resolution OGCMs might not cause significant issues. However, as the horizontal resolution of OGCMs increases, mesoscale or even submesoscale dynamics with multiscale interactions enter the model grid-scale regime, and local features should be considered when parameterizing subgrid-scale effects. Within the LES framework, subgrid-scale stress and flux terms that include local interactions can be fully expressed, thereby improving the simulation results. Second, the LES framework can also explicitly involve the stationary eddying effect as a supplement to traditional schemes that mainly focus on the transient eddying effect related to instabilities (Khani and Dawson, 2023; Xie et al., 2023). In addition, constructing parameterization

schemes that combine LES with machine learning has become a frontier field in developing OGCMs (Bolton and Zanna, 2019; Zanna and Bolton, 2020; Guillaumin and Zanna, 2021; Frezat et al., 2022; Srinivasan et al., 2024).

There are two types of tests for examining the performance of LES models (i.e. parameterization schemes in oceanography): *a priori* and *a posteriori* tests (Meneveau, 1994; Moser et al., 2021). In an *a priori* test, direct numerical simulation (DNS) for a specific flow is required first. Then, we perform the scale separation of DNS data through coarse-graining methods (e.g., spatial filtering). We regard the filtered field as an approximation of the coarser-resolution model output, and we directly diagnose the "true" subgrid-scale terms following their definition. Finally, we reconstruct the subgrid-scale terms by substituting the filtered field into the LES model. By investigating the performance of the reconstructed and the "true" subgrid-scale terms under given metrics (e.g., spatial correlations and the energy transfer rate), we can find out the properties of the LES model or parameterization scheme. In an *a posteriori* test, we embed the given LES scheme into a lower-resolution numerical model, run simulations, and test parameter sensitivity if needed. At this point, the macroscopic features of the corresponding flow from DNS data serve as the benchmark for evaluating the lower-resolution simulation results when applying the LES model. For both *a priori* and *a posteriori* tests, DNS data for the studied flow are essential. Therefore, when introducing LES methods into ocean mesoscale parameterization, it is necessary to generate DNS datasets for mesoscale processes, thereby facilitating the systematic work of developing, testing, and implementing any LES schemes into OGCMs.

In the literature of computational fluid dynamics, DNS requires the numerical model resolution to be at least close to the Kolmogorov scale (Moin and Mahesh, 1998; Pope, 2000; Kaneda and Ishihara, 2006; Alfonsi, 2011). It is the scale at which molecular viscosity becomes important (Pope, 2000; Vallis, 2017). However, the classical definition of DNS is inapplicable in the context of implementing LES methods into OGCMs. Carrying out simulations with resolution close to the Kolmogorov scale in OGCMs is not feasible in the foreseeable future, and it is also unnecessary to adopt such high resolution for merely simulating mesoscale motions. For oceanic mesoscale flow, the dynamically indicative scale is the first baroclinic deformation radius. The radius is in the range of approximately $10 \sim 40$ km in mid-latitude oceans (e.g., the Southern Ocean) (Chelton et al., 1998; LaCasce and Groeskamp, 2020), necessitating a horizontal resolution of at least $1/30°$ to resolve it explicitly (Hallberg, 2013; Marques et al., 2022). In addition, OGCMs often adopt dissipative schemes near the grid scale to ensure numerical stability. If the scale at which the dissipative scheme plays a significant role cannot be well separated from the first baroclinic deformation radius, then the intrinsic mesoscale dynamics would be contaminated artificially. Some works (e.g., Graham and Ringler, 2013; Radko and Kamenkovich, 2017) use the term DNS in the context of ocean mesoscale dynamics, which might lead to misunderstandings. Therefore, we hereby explicitly propose the concept of mesoscale ocean direct numerical simulation (MODNS). A qualified MODNS dataset not only requires the model grid to explicitly resolve the first baroclinic deformation radius but also demands that the affected scales by the dissipative scheme employed significantly smaller than the radius, making it the benchmark for *a priori* and *a posteriori* tests of LES schemes (or more specifically, MOLES methods) into OGCMs.

To highlight the oceanic mesoscale dynamics in the simulation while reducing computational and storage costs, we develop an idealized Southern Ocean model (ISOM 1.0) and conduct fully eddy-resolving experiments to generate a type of MODNS dataset. Though it is fundamental for the simulation to conform to the realistic Southern Ocean in terms of basic dynamical

features (i.e., quantitatively consistent with observed ACC transport value to provide a reasonable background flow for eddying processes), we emphasize that the focus of the simulation should be on controlling the dynamics of the idealized model rather than on precise comparisons with observations or realistic model results. We hope that the model can describe processes most closely associated with the mesoscale in the Southern Ocean, including mesoscale motions (mesoscale eddies and meandering jets), large-scale background processes (stratification and eastward transport similar to that in the realistic Southern Ocean), eddy-eddy interactions, eddy-jet interactions, large-scale topographic effects, and mesoscale-submesoscale interactions. ISOM enables us to achieve a type of MODNS in an idealized Southern Ocean with topography, thereby providing reliable supporting data for the design, testing, and application of any potential LES-related mesoscale parameterization schemes and the theoretical exploration of the dynamics.

We introduce the design philosophy and implementation of ISOM 1.0. We verify that the oceanic mesoscale regime is fully resolved and barely contaminated by the dissipative schemes in the highest-resolution simulation. Therefore, the simulation can serve as a type of MODNS dataset with vigorous mesoscale activities. We also offer vivid examples of multiscale eddy interactions, especially the eddy-jet interaction, to intuitively demonstrate the capability of MODNS to capture all the mesoscale-related processes. In addition, we conduct multipassive tracer experiments and explore the principles for setting the initial field of passive tracers to offer technical references for relevant works.

## 2 Model description

### 2.1 Model equations and configurations

We establish ISOM 1.0 using the Massachusetts Institute of Technology general circulation model (MITgcm; Marshall et al., 1997). We refer to and improve upon the case of Southern Ocean Reentrant Channel Example in the MITgcm manual (Adcroft et al., 2021) that is closest to our needs, as well as similar idealized works (e.g., Abernathey et al., 2011; Bischoff and Thompson, 2014), ultimately achieving ISOM 1.0 with topography and intermediate complexity. Similar to these works, we consider a hydrostatic, incompressible Boussinesq fluid on the $\beta$-plane, with an implicitly linearized free surface and a linearized equation of state (only potential temperature, no salinity). We employ the Cartesian coordinate, and then the governing equations (without specifying concrete parameterizations) are as follows:

$$\frac{Du}{Dt} - fv + \frac{1}{\rho_c}\frac{\partial p'}{\partial x} + \nabla_h \cdot (-A_h \nabla_h u) + \frac{\partial}{\partial z}\left(-A_v \frac{\partial u}{\partial z}\right) = \mathcal{F}_u, \tag{1}$$

$$\frac{Dv}{Dt} + fu + \frac{1}{\rho_c}\frac{\partial p'}{\partial y} + \nabla_h \cdot (-A_h \nabla_h v) + \frac{\partial}{\partial z}\left(-A_v \frac{\partial v}{\partial z}\right) = \mathcal{F}_v, \tag{2}$$

$$\frac{\partial \eta}{\partial t} + \nabla_h \cdot (H\widehat{\boldsymbol{u}}) = 0, \tag{3}$$

$$\frac{D\theta}{Dt} + \nabla_h \cdot (-\kappa_h \nabla_h \theta) + \frac{\partial}{\partial z}\left(-\kappa_v \frac{\partial \theta}{\partial z}\right) = \mathcal{F}_\theta, \tag{4}$$

$$p' = g\rho_c \eta + \int_z^0 g\rho' dz. \tag{5}$$

Here, $u$ and $v$ are the $x$ and $y$ components of the velocity vector, $\eta$ is the free surface height, $\theta$ is the potential temperature, and $p'$ is the pressure field. For more details, please refer to the MITgcm manual.

The forcing terms of the horizontal momentum equation $\mathcal{F}_u$ and $\mathcal{F}_v$ include the steady zonal surface wind stress and the quadratic bottom drag. The wind stress is set as follows:

$$\tau_s(y) = \tau_0 \sin(\pi y / L_y), \tag{6}$$

with $\tau_0 = 0.2 Nm^{-2}$. The dimensionless coefficient of quadratic bottom drag $C_d$ is 0.01.

The forcing term of the potential temperature equation $\mathcal{F}_\theta$ includes relaxation to a prescribed surface temperature profile and
a sponge layer at the northern side of the domain. The specified sea surface temperature profile increases linearly from $0°$C in the south to $20°$C in the north, with a relaxation time scale of 30 days (except in the sponge layer region). The sponge layer, or a three-dimensional subdomain where restoring boundary conditions are applied, is confined within 160 km at the northern side of the domain, and the potential temperature is relaxed to the following profile:

$$\theta_p(y,z) = [\theta_s(y) - \theta_b]\left(e^{z/h_0} - e^{-H/h_0}\right) / \left(1 - e^{-H/h_0}\right). \tag{7}$$

$\theta_s$ is the prescribed surface temperature profile that varies linearly and meridionally, $\theta_b$ is the bottom temperature set to $0°$C. The depth of the domain $H$ is 4000 m, and the scaling height $h_0$ is 1200 m (note that the z-coordinate origin is at the surface and $z$ in the equation is minus). The setup can well represent the stratification on the northern side of the Antarctic Circumpolar Current (ACC; Abernathey et al., 2011). The relaxation time scale within 80 km at the northernmost part of the sponge layer is 7 days, and from 80 to 160 km is 14 days.

We use the combination of the horizontal Laplacian viscosity $A_h$ and biharmonic viscosity $A_4$ for the closure of the fourth term in horizontal momentum equations. We set the background vertical Laplacian viscosity $A_v = 3 \times 10^{-4}$ m s$^{-2}$ for the fifth term in horizontal momentum equations. We also set the background vertical Laplacian diffusivity $\kappa_v$ and use no horizontal diffusivity $\kappa_h$ in the potential temperature equation. In addition, we use the nonlocal K-Profile Parameterization (KPP) scheme (Large et al., 1994) for vertical mixing to generate the mixed layer. Tables 1 and 2 offer the setting values for all the relevant
parameters.

## 2.2   Model bathymetry

The computational domain is a 18000 km $\times$ 3000 km channel with a prescribed topography and zonal periodic boundary conditions (Fig. 1). The domain depth is 4000 m, with 75 vertical levels, and the vertical grid spacing increases from 2 m

**Table 1.** Basic parameters of the idealized Southern Ocean simulation.

| Symbol | Value | Description |
|---|---|---|
| $L_x, L_y$ | 18000 km, 3000 km | Domain size |
| $H$ | 4000 m | Domain depth |
| $\Delta z$ | 2 - 125 m | Vertical grid spacing |
| $L_{sponge}$ | 160 km | Sponge layer size |
| $\tau_{sponge}$ | 7 days | Shortest Sponge layer relaxation time scale |
| $\lambda$ | 30 days | Surface temperature relaxation time scale (outside the sponge layer) |
| $f_0$ | $-1 \times 10^{-4} \text{s}^{-1}$ | Reference Coriolis parameter |
| $\beta$ | $1 \times 10^{-11} \text{m}^{-1}\text{s}^{-1}$ | Meridional gradient of Coriolis parameter |
| $g$ | 9.81 m s$^{-2}$ | gravitational acceleration |
| $\tau_0$ | 0.2 N m$^{-2}$ | Wind stress magnitude |
| $C_d$ | $1 \times 10^{-2}$ | Quadratic bottom drag parameter |
| $\rho_c$ | 1035 kg m$^{-3}$ | Reference density |
| $\alpha$ | $2 \times 10^{-4}\text{K}^{-1}$ | Linear thermal expansion coefficient |
| $\kappa_v$ | $5 \times 10^{-6}$ m s$^{-2}$ | Vertical diffusivity |
| $\kappa_h$ | 0 | Horizontal diffusivity |
| $A_v$ | $3 \times 10^{-4}$ m s$^{-2}$ | Vertical viscosity |

**Table 2.** Parameters of simulations with different horizontal resolution at their statistical steady state.

| Symbol | Value-1 | Value-2 | Value-3 | Description |
|---|---|---|---|---|
| $\Delta x, \Delta y$ | 2 km | 4 km | 8 km | Horizontal grid spacing |
| $A_h$ | 2 | 10 | 50 | Horizontal Laplacian viscosity(m s$^{-2}$) |
| $A_4$ | $1 \times 10^8$ | $5 \times 10^9$ | $1 \times 10^{10}$ | Horizontal biharmonic viscosity(m$^4$ s$^{-1}$) |
| $\Delta t$ | 80 s | 250 s | 300 s | time step |

at the surface to 125 m at the bottom. The channel mimics the hemispheric Southern Ocean that spans from the west of the
Drake Passage to the east of the Kerguelen Plateau. This region has a highly complex topography and contains most of the
iconic bathymetric features of the Southern Ocean. These features exert profound impacts on the holistic Southern Ocean flow.
Using these iconic features enables the idealized model to preserve the complicated topographic effects on oceanic mesoscale
processes. Although we could fill the entire domain with realistic topography, it would cause the bathymetric features to be too

close to each other and excessively suppress the development of flow. Moreover, reducing computational and storage costs is
another reason for imitating only half rather than the entire Southern Ocean.

We adopt five types of topography in the domain (Fig. 1a, b). They are described in detail from left to right as follows.

(1) The first type is an idealized Drake Passage that is a passage from x = 800 km to x = 3200 km. It smoothly narrows from
the sides to form a 1600 km × 600 km rectangular subchannel. The four corners of the passage are 1/4 circular arcs with a
radius of 400 km. The bottom topography within the passage rises from -4000 m to -2500 m within the range of x = 800 km to
x = 2600 km and descends from -2500 m to -4000 m within the range of x = 2600 km to x = 3200 km.

The design helps the idealized model qualitatively reproduce the flow characteristics near the realistic Drake Passage despite
its high simplification (Fig. 1c, d). In our early design, we tested the passage with a flat bottom or the semicircular ridge similar
to Marques et al. (2022), but found that they somehow suppress eddy activity near the passage.

We also tested the passage's depth and width in the early spin-up stage. Deepening the passage (e.g., changing the highest
point to -3000 m) and widening the passage (e.g., increasing the north-south span to 750km) can significantly increase the
ACC transport value to over 200 Sv. Different observations and simulations (e.g., Cunningham et al., 2003; Park et al., 2009;
Donohue et al., 2016; Xu et al., 2020; Artana et al., 2021a) yield varying values for ACC transport, but generally, they fall
within the range of 130 to 180 Sv, which is also consistent with our calculation using Southern Ocean State Estimate (SOSE).
To keep the ACC transport of ISOM within the reasonable range, we decided to make the design described above so that
the value in the early spin-up stage is around 170 Sv. As the simulation continued, the ACC transport slowly decreased and
reached an equilibrium state of around 145 Sv, which is still within the reasonable range after 30 model years of integration.
The time-averaged ACC transport can remain at the same level after horizontal resolution increases (Figs. 2a and 3a).

In addition, we also found in the early design (Fig. S3 and Tables S1 and S2 in the supplementary material) that when the
depth of the passage is too shallow (e.g., -1000 m), ACC transport is highly sensitive to the bottom drag coefficient. Reducing
(increasing) the coefficient can significantly increase (decrease) ACC transport. In our current design (i.e., -2500 m), ACC
transport is no longer sensitive to the bottom drag coefficient.

(2) The second type is a tilted mid-ocean ridge (ridge-1) with zonal and meridional slopes downstream of the idealized Drake
Passage with a zonal width $w = 3500$ km and height $h_1 = 800$ m. The ridge is confined within a parallelogram with vertices
at (x, y) = (3500 km, 3000 km), (7000 km, 3000 km), (4500 km, 0), and (8000 km, 0). In the zonal cross-section, the ridge is
composed of piecewise cubic functions with height $h$ expressed as follows:

$$h(x) = \begin{cases} -\frac{16h_1}{w^3}x^3 - \frac{12h_1}{w^2}x^2 + h_1, & -\frac{w}{2} < x < 0, \\ \frac{16h_1}{w^3}x^3 - \frac{12h_1}{w^2}x^2 + h_1, & 0 < x < \frac{w}{2}. \end{cases} \tag{8}$$

The ACC flow intensity in the realistic Atlantic sector is weaker than in the vicinity of the Drake Passage, Agulhas region,
and Kergulen Plateau region. To maintain this feature in ISOM, we set a low mid-ocean ridge with an undulation (800 m)
weaker than the realistic ridge in the South Atlantic and our early design (1500 m, Fig. S3 in the supplementary material).
We explain that the idealized mid-ocean ridge is a continuous and complete topography in the north-south orientation, which
has a more concentrated stimulation effect on the activity of eddies and jets compared to the realistic ridge with fractures
and breakpoints. The low mid-ocean ridge avoids its topographic effect from excessively enhancing eddy activities, thereby

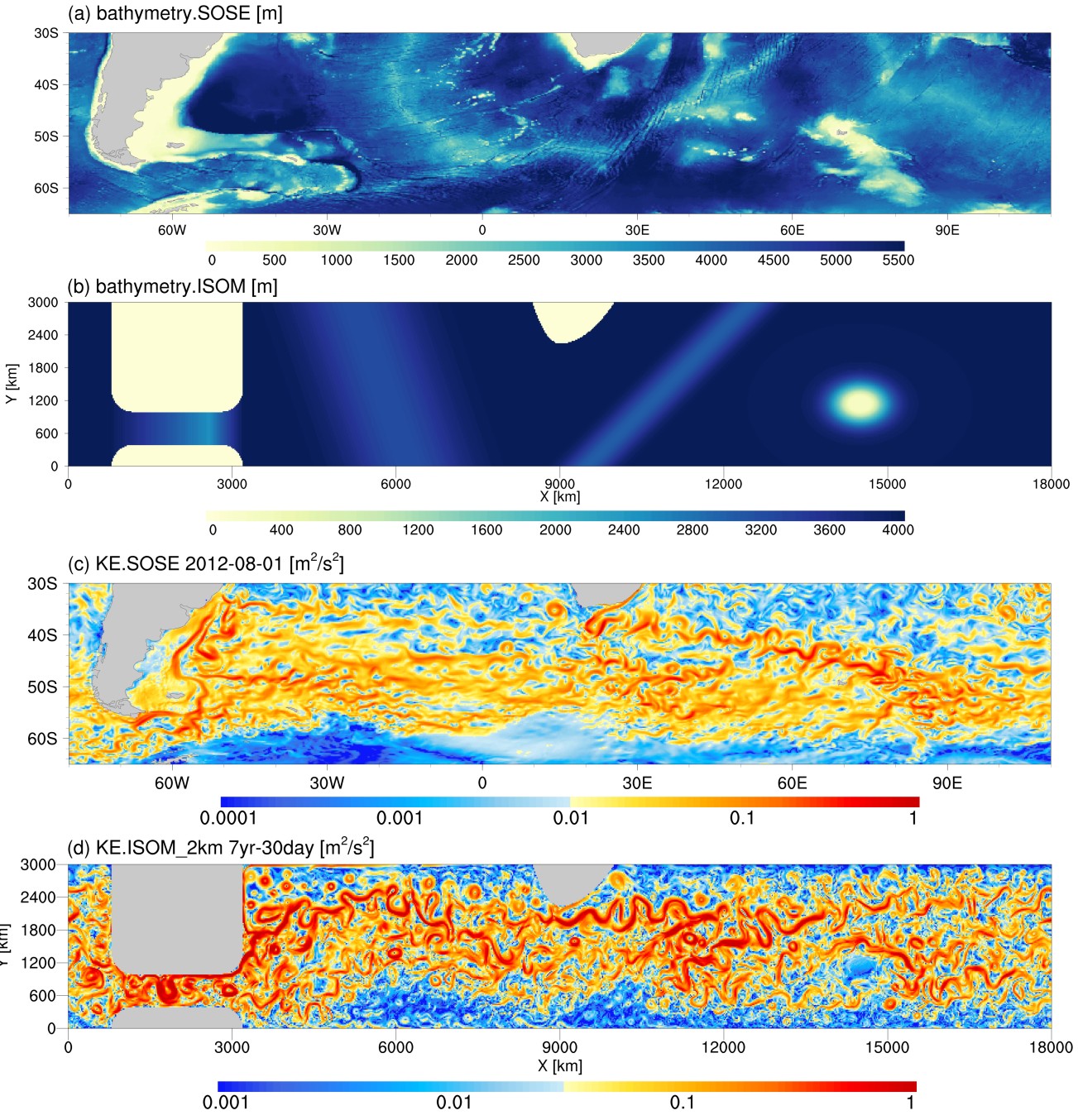

**Figure 1.** The bathymetry of (a) Southern Ocean State Estimate (SOSE; Mazloff et al., 2010; Verdy and Mazloff, 2017) and (b) ISOM 1.0. The snapshots of the surface kinetic energy of (c) SOSE and (d) 2-km-resolution ISOM simulation.

avoiding intense downstream flushing and improving the results of the retroflection feature and eddy-shedding process in the Agulhas region.

(3) The third type is an idealized African continent. We notice the asymmetry of the southern African and define the profile with the quadratic function $y(x) = \frac{3}{1000}(x - x_0)^2 + y_0$ from x = 8500 km to x = 9000 km and $y(x) = \frac{3}{4000}(x - x_0)^2 + y_0$ from x = 9000 km to x = 10000 km and the vertex $(x_0, y_0)$ = (9000 km, 2250 km). Compared to the early version with a symmetric design (Fig. S3), the introduction of asymmetry enhances the imitation of Agulhas retroflection feature.

    (4) The fourth type is another mid-ocean ridge (ridge-2) with opposite tilt direction and a zonal width $w$ = 1500 km and
190 height $h_1$ = 1000 m. The ridge is confined within a parallelogram with vertices at (x, y) = (11750 km, 3000 km), (13250 km, 3000 km), (8750 km, 0), and (10250 km, 0). In the zonal cross-section, the expression of height $h$ is the same as Eq. (8).

    (5) The fifth type is an idealized Kergulen Plateau. We set a large-scale elliptical Gaussian plateau centered on $(x_m, y_m)$ = (14500 km, 1150 km), and its height $h$ is expressed as follows:

$$h(x, y) = h_2 \exp\left[-\frac{(x - x_m)^2}{2\sigma_x^2} - \frac{(y - y_m)^2}{2\sigma_y^2}\right]. \tag{9}$$

$\sigma_x$ is taken as 350 km, $\sigma_y$ is 250 km, and $h_2$ is 4500 m. We further limit $h_2$ to be no higher than 3800 m to form an elliptical flat plateau underwater.

    Though the model bathymetry is highly idealized, ISOM qualitatively reproduces the major flow characteristics of the Southern Ocean (Fig. 1c, d), such as intense eddy activity, enhanced currents downstream of large-scale topographic features, and the eastward extension of the current from southern Africa, and successfully simulates energetic mesoscale and submesoscale
phenomena and the vivid multiscale eddy-eddy and eddy-jet interactions (see Section 3). By observing the differences between the simulated results and the realistic Southern Ocean and referring to other relevant works, we can gain experience and lessons for potential optimization of the model. We share the following insights and hope they can be helpful to any researchers who conduct similar works in the future.

    (1) Other idealized models of similar complexity, such as Neverworld in Khani et al. (2019) and Neverworld-2 in Marques
et al. (2022), often apply a steep slope transition from the landmass to the domain depth to mimic the continental shelf. We find in tests that this leads to the formation of intense currents driven by the topographic $\beta$ effect near the side boundaries. Unfortunately, these currents, especially near the southern boundary, interfere with the simulation of other processes in ISOM. Marques et al. (2022) mentioned their application of significant lateral dissipation. It might be a way to weaken the overly abundant boundary currents. However, the goal of our simulation is to obtain the MODNS dataset. To avoid directly con-
taminating mesoscale dynamics, we are not inclined to use excessive viscous dissipation in fully eddy-resolving simulations. Therefore, we abandoned the design of continental shelves. Nevertheless, after we completed the simulation, we found that the result of the near-shore Agulhas current was weaker than in SOSE. Thus, we recommend that future researchers add a moderate topographic slope around the African continent if necessary.

    (2) The complicated bathymetry near the South American continent has a decisive influence on the surrounding flow, especially
the presence of the Malvinas Islands shelf, which leads to a narrow and extensive boundary current (Artana et al., 2021b).

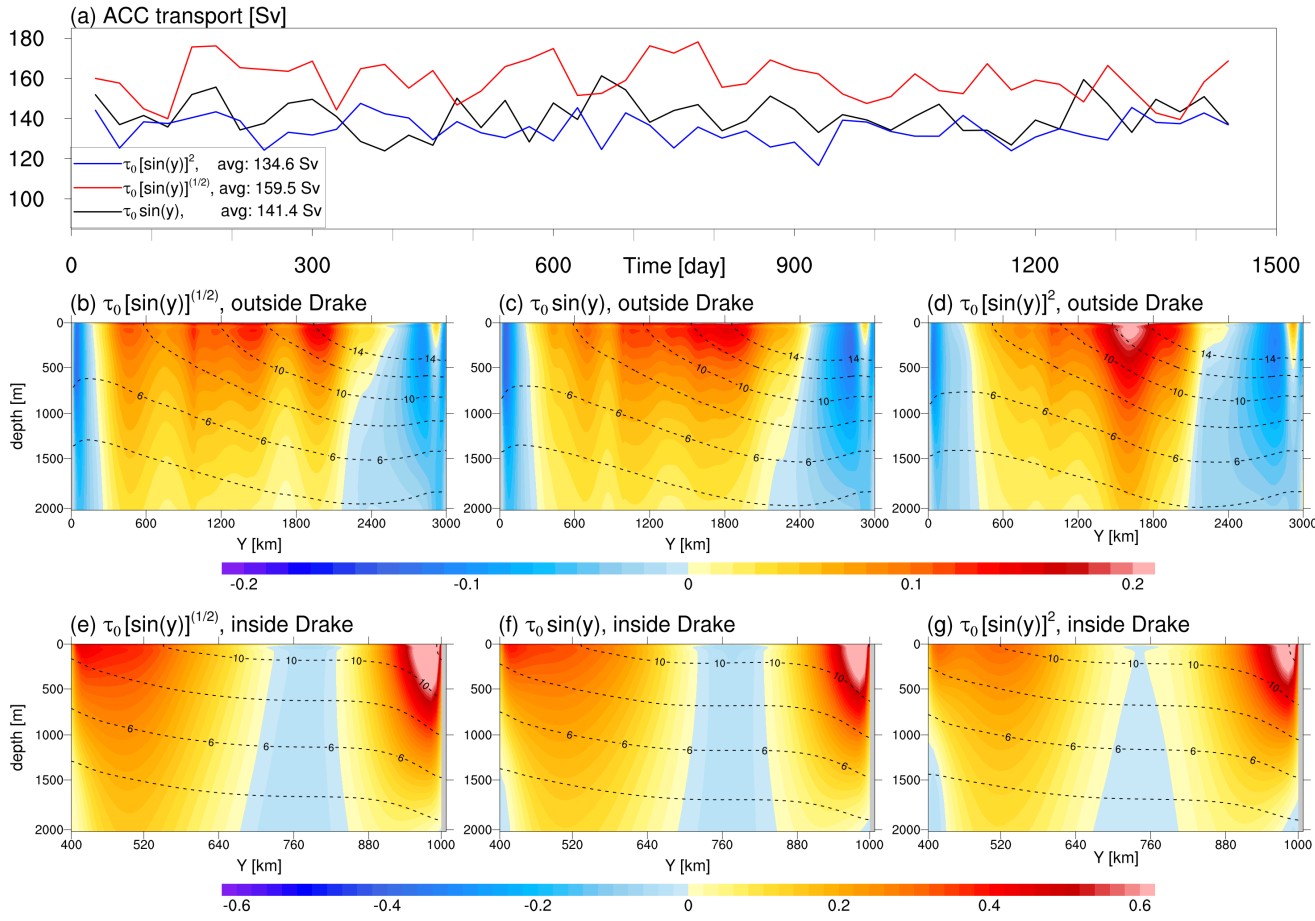

**Figure 2.** (a) The monthly-averaged ACC transport under three wind stress profiles. (b-d) are zonal-averaged zonal velocity (shading) and zonal-averaged potential temperature (contours) outside the idealized Drake passage under three wind stress profiles. (e-g) are as (b-d) but inside the idealized Drake passage.

ISOM 1.0 prioritizes imitating the Drake Passage. If researchers conduct similar works in the future, we suggest optimizing the topographical expression along the South American continent.

(3) Previous studies (e.g., Speich et al., 2006; Lutjeharms and Van Ballegooyen, 1984; Lutjeharms, 2007) reveal that the topography near the Agulhas Retroflection region, such as the Agulhas Bank (the continental shelf extending southwest from the African continent) and the Agulhas plateau (a large-scale seamount offshore to southeastern Africa), plays a crucial role in controlling the flow state of eddies and jets. Altering topographic features, such as the slope of the continental shelf and the degree of topographic undulations, substantially influence the flow path, eddy-shedding process, and cross-basin transport. ISOM 1.0 violently simplifies the bathymetric environment near South Africa. We recommend adding the topographic details

around the continent if future works want to improve the simulation of the retroflection characteristics and the eddy-shedding

process of the Agulhas Rings.

(4) Using a sufficiently wide north-south domain span allows for a more reasonable arrangement of the relative positions of topographic features. In the early design (Fig. S3), we used a north-south span of 2400 km, which could not realize a correct relative position between the African continent and the Drake Passage (i.e., if we set the African continent far north, the flow would be directly and unwantedly affected by the sponge layer), thus distorting the holistic pattern of flow. The current

design, with a north-south span of 3000 km, offers us more space to optimize the arrangement of iconic bathymetric features, dramatically improving the resemblance of ISOM results to the realistic situation. In addition, a sufficiently long east-west span is also conducive to the full development of flow. Researchers may enhance the resemblance by further enlarging the domain and optimizing topographic arrangement based on this manuscript. However, one has to consider the compromise and balance between the extra computational costs and experimental aims. For example, our early design uses a domain of 14400 km $\times$

2400 km with 40 vertical levels (Fig. S3), while the current design uses a domain of 18000 km $\times$ 3000 km with 75 levels. The overall computational load increased by nearly three times, which exerted a significant burden during the simulation.

## 2.3  Sensitivity of wind stress profile

In this section, we discuss the sensitivity of the flow and stratification to wind stress settings. From Eq. (6), wind stress consists of two parts: the amplitude $\tau_0$ and the profile form. Many previous studies (e.g., Abernathey and Cessi, 2014; Balwada et al.,

2018; Khani et al., 2019; Marques et al., 2022) set $\tau_0 = 0.2\ Nm^{-2}$ consistent with the observation (Chaudhuri et al., 2013; Abernathey et al., 2011) and the effect of amplitude $\tau_0$ has been extensively studied (Tansley and Marshall, 2001; Bischoff and Thompson, 2014; Youngs et al., 2017). Thus, we focus on the effect of the wind stress profile. Idealized simulations often employ the profile related to the sinusoid function. For example, Abernathey et al. (2011) used the standard sinusoid form, while Balwada et al. (2018) adopted the quadratic form in sinusoid. We tested half-power ($\tau_0 \sin^{1/2}[\pi y/L_y]$), standard

($\tau_0 \sin[\pi y/L_y]$), and quadratic ($\tau_0 \sin^2[\pi y/L_y]$) forms in sinusoid because the wind stress profile setting for the idealized simulations is unlikely to exceed this range. Otherwise, it would deviate significantly from the observed situation. We integrated four model years under three wind stress profiles, starting from the last moment of the 51st model year of the 8-km simulation.

Figure 2a shows the time series of ACC transport under three wind stress profile. The response of ACC transport to the wind stress profile is quick (compared to the whole spin-up process). In less than one model year, the ACC transport under the

half power of sinusoidal wind stress forcing is prominently stronger than others and basically reaches a new statistical steady state. The higher the power exponent of the sinusoidal function, the smaller the overall ACC transport. The last two-year time-averaged ACC transport is 134.6 Sv, 141.4 Sv, and 159.5 Sv for quadratic, standard, and half-power forms, respectively. The fundamental reason for this result is that the higher the power exponent of the sinusoidal wind stress profile, the sharper it decays to both sides (north and south). When it reaches the latitude (y-coordinate) where the Drake Passage is, the magnitude

of the wind stress with the high-power form is smaller than that of the low-power form.

We further demonstrate the time- and zonal-averaged potential temperature and zonal velocity under three wind stress profiles (Fig. 2b-g). Since the northern boundary geostrophic flow of the Drake Passage is very energetic with a steep isothermal

slope, if we zonally average the entire channel, a discontinuity appears on the figure. But in reality, the fields are continuous with large meridional gradients. Therefore, to avoid the misleading visual illusion and to show in detail the response of flow and stratification to the wind stress profile, we divide the entire channel into two parts for zonal average. One is the rectangular sub-channel within the Drake Passage with x ∈ [1200 km, 2800 km] and y ∈ [400 km, 600 km]. The other is the rest of the domain.

Outside the Drake Passage (Fig. 2b, c, d), as the power exponent of the sinusoidal wind stress increases, the zonal flow structure becomes more compact. An intense jet core condenses around the domain center under the quadratic form of forcing (Fig. 2d). In the Passage (Fig. 2e, f, g), the northern boundary flow is more vigorous than the southern boundary flow. As the power exponent of the sinusoidal wind stress decreases, the boundary flows become more energetic and penetrate deeper (Fig. 2e).

In summary, if one wants to study oceanic processes on a concentrated zonal jet like Balwada et al. (2018), a high-power sinusoidal wind stress profile would be a good choice. If one wants to generate a holistically stronger ACC (especially within the Drake Passage), we recommend the profile with a low-power sinusoidal function. We choose the standard form in the manuscript to maintain consistency with previous relevant studies and control the ACC transport value in the reasonable range.

## 2.4 Implementation of MODNS

The fundamental requirement for MODNS is that the horizontal resolution can explicitly resolve the first baroclinic deformation radius. Under the model configuration, it is basically greater than 15 km. If we follow the experience that the model discretization can adequately represent processes exceeding five times the grid spacing, then the horizontal resolution required for MODNS should reach at least 3 km. Additionally, Marques et al. (2022) found that when the horizontal resolution reaches $1/32°$ in Neverworld-2, the mesoscale model performance converges, meaning that the resolution is sufficient to fully resolve mesoscale in their model. Based on these considerations, we conduct a simulation with an even finer horizontal resolution of 2 km to achieve a certain type of MODNS.

Directly running the 2-km simulation (e.g., spin up from rest) is costly. Therefore, our spin-up strategy is as follows: (1) Integrate an 8-km simulation from rest for 51 model years to reach the quasi-equilibrium state. (2) Interpolate the final output of the 8-km simulation as the initial field for the 4-km simulation and integrate for 16 model years to reach the corresponding quasi-equilibrium state. (3) Interpolate the final output of the 4-km simulation as the initial value field for the 2-km simulation and integrate for several model years.

We employ the 3rd-order direct space-time (DST) flux limiter scheme (the MITgcm advection scheme code is 33) for all simulations. Since the 2-km simulation has already been submesoscale-permitting, we retain the 3-day average output from the 7th model year and daily instantaneous fields from the first two months in the 8th model year in the 2-km simulation to demonstrate the model performance for eddying processes of different scales, including mesoscale and submesoscale.

Figure 3 displays the ACC transport series and Rossby number (the vertical component of relative vorticity normalized by local Coriolis parameter, see Section 3.3 for detailed definition) snapshots at three horizontal resolutions. The ACC transport has reached quasi-steady state after integrating the 8-km simulation into 48 model years (Fig. 3a). After improving the hor-

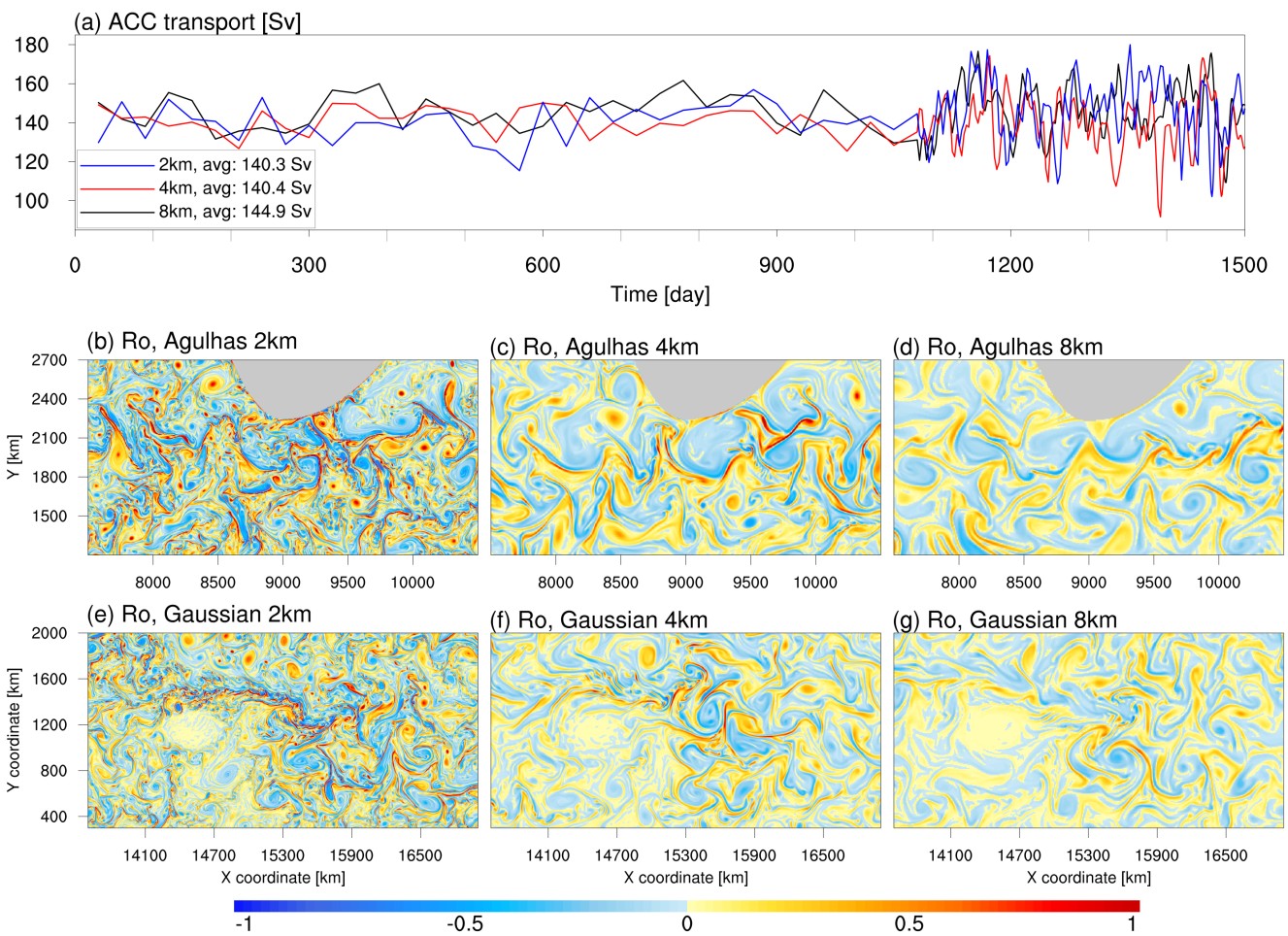

**Figure 3.** (a) The time series of ACC transport spans 1500 model days. The results are monthly averages for the first three model years, 3-day averages for the fourth model year, and daily snapshots for the last 60 model days. The blue line for the 2-km simulation starts from the 4th model year. The red line for the 4-km simulation starts from the 13th model year. The black line for the 8-km simulation starts from the 48th model year. The ACC transport in the 8-km simulation has reached a statistically steady state in the 48th model year. After refining the horizontal resolution, the ACC transport can also maintain a statistically steady state close to the result of the 8-km simulation. (b-d) for the Rossby number (see Section 3.3 for detailed definition) in the idealized Agulhas region for simulations with different horizontal resolutions. (e-g) as (b-d) but for the Gaussian plateau region.

izontal resolution, in the last few years of the 4-km and 2-km simulations, the ACC transport keeps in equilibrium with no substantial change compared to the results at 8 km. The magnitude of the Rossby number considerably increases with finer horizontal resolution (Fig. 3b-g). It indicates that the model can explicitly describe more submesoscale phenomena, especially those related to meandering jets, filamentation, and multiscale interactions. We will discuss these in detail in Section 3.3.

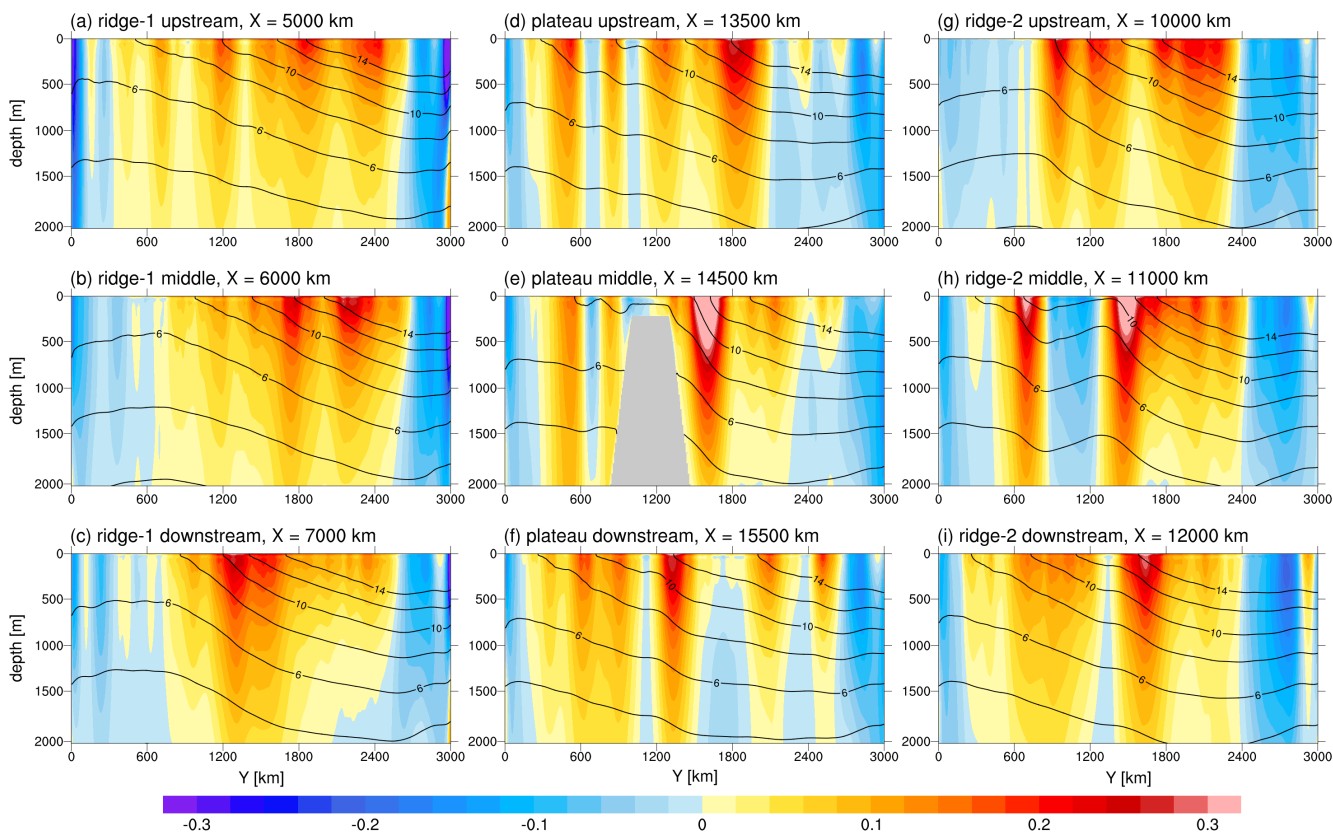

**Figure 4.** The meridional cross sections of 3-year-averaged potential temperature (contours) and zonal velocity (shading) of the 2-km simulation. (a-c) for the western ridge from upstream to downstream, (d-f) for the Gaussian plateau, and (g-i) for the eastern ridge.

## 3    Results

In this section, we present the stratification, jets, kinetic energy and enstrophy spectrum, and the eddying field from the 2-km simulation to examine the model performance for processes relevant to oceanic mesoscale dynamics.

### 3.1    Stratification and jets

We show the time-averaged temperature and zonal velocity component on the meridional cross-sections from upstream to downstream of the two mid-ocean ridges and Gaussian plateau (Fig. 4) and demonstrate that the model can capture the impact of large-scale topography on the stratification and zonal flow state.

Upstream of ridge-1 (the western ridge), the zonal flow exhibits a multi-branched state, accompanied by gentle isotherm slopes (Fig. 4a). The flow drifts to the north on the ridge and gets intensified (Fig. 4b). Then, the flow drifts to the south further

downstream and becomes more compact (Fig. 4c) with vigorous eddy activities (Fig. 1d) and potentially high eddy diffusivity

(Abernathey and Cessi, 2014), forming a distinct jet core corresponding to steeper isotherm slopes under the geostrophic constraint.

As for the flow in the Gaussian plateau region, multiple branches upstream of the topography exist (Fig. 4d), with the jet core at y = 1900 km. When the flow passes the plateau (Fig. 4e), it is steered along the northern side and forms a single energetic jet core with a steep isotherm slope. Downstream of the plateau (Fig. 4f), the zonal flow scatters into several branches again, and the jet core moves further south to y = 1300 km, where the eddy kinetic energy (Fig. 1d) and mixing induced by stationary eddy would be drastic (Bischoff and Thompson, 2014; Xie et al., 2023).

Ridge-2 (the eastern ridge) has a more complicated bathymetric environment. Its upstream section (Fig. 4g) passes through the Agulhas retroflection zone. Thus, the northern eastward jet reflects the Agulhas retroflection current. When crossing the ridge (Fig. 4h), the strong topographic slope remarkably strengthens the jets. Due to its narrow spatial scale, the jets do not completely merge when crossing topography and downstream (Fig. 4i).

The above phenomena are qualitatively consistent with previous studies that used two-layer quasigeostrophic models with bottom slopes or topography (Tansley and Marshall, 2001; Thompson, 2010; Chen et al., 2015; Khatri and Berloff, 2018), idealized mid-ocean ridge experiments (Abernathey and Cessi, 2014; Youngs et al., 2017), idealized Gaussian plateau experiments (Bischoff and Thompson, 2014), realistic Southern Ocean topography simulations (Thompson and Naveira-Garabato, 2014), Southern Ocean reanalysis data (Lu and Speer, 2010; Abernathey and Cessi, 2014), laboratory experiments (Rhines, 2006), and observational data (Orsi et al., 1995; Thompson and Sallee, 2012; Chapman et al., 2020). This indicates that ISOM 1.0 is capable of describing the large-scale background processes that are closely associated with mesoscale phenomena in the Southern Ocean.

## 3.2 EKE and enstrophy spectrum

To better extract eddy signals, we take 1024 km zonal segments at given locations and compute the EKE spectrum from the meridional component of eddy velocity, similar to Marques et al. (2022). We define the eddy velocity by subtracting the temporal and zonal segment-mean velocity. Figure 5a-c shows the EKE spectrum of three-day averaged model output from one model year with three horizontal resolutions. The high-frequency processes (e.g., submesoscale) are thus filtered out, and the output highlights the model performance in simulating mesoscale processes. In regions with active mesoscale eddies, including the Drake Passage, downstream of the two mid-ocean ridges, the Agulhas region, and downstream of the Gaussian plateau, the results consistently exhibit a -3 spectral slope for most of the spectrum of all simulations. As the resolution improves, the range of -3 spectral slope expands. For the 2-km simulation (Fig. 5c), the range of the -3 spectral slope (> 8 km) completely covers the deformation radius scale (> 15 km), which also means that the dissipative schemes barely contaminate the modeled mesoscale process. This result suggests that the model can fully describes oceanic mesoscale motions that can be deemed quasigeostrophic turbulence and theoretically possesses a $k^{-3}$ ($k$ is wavenumber) kinetic energy spectrum (Charney, 1971; Fu and Morrow, 2013; Vallis, 2017).

Figure 5d-f displays the EKE spectrum computed from the daily instantaneous fields of sixty model days of simulations with three resolutions. The 8-km and 4-km simulations maintain a spectral slope -3 related to mesoscale processes over most

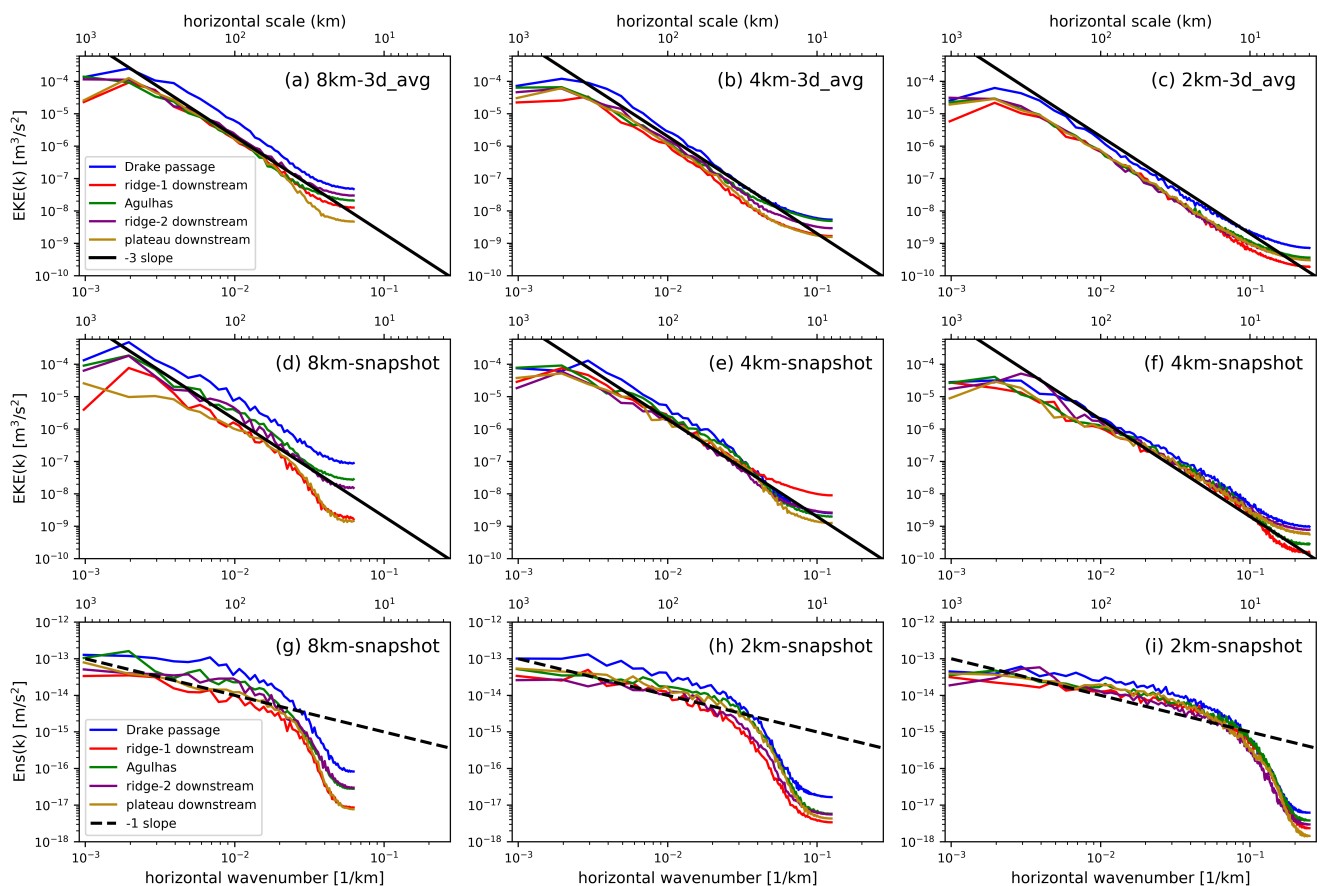

**Figure 5.** The averaged spectral density function of surface EKE and enstrophy (relative vorticity variance) for simulations with different resolutions. (a-c) show the surface EKE spectrum using the 3-day-averaged output of one model year. (d-f) show surface EKE spectrum using daily snapshots of sixty model days. (g-i) show the surface enstrophy spectrum using daily snapshots of sixty model days. Lines with different colors represent different sampling positions or reference slopes. The eddy velocity is defined by subtracting the time and zonal average of a 1024-km zonal segment of meridional velocity in each position.

scale ranges (Fig. 5d, e). The 2-km simulation (Fig. 5f) shows a slight uplift at scales below 40 km, resulting in a spectral slope weaker than -3. The spatial distribution and statistics of sea surface variables (Fig. 3 and Tables 3 and 4) confirm that the 2-km simulation models remarkably more submesoscale signals compared to the lower resolution simulations. Thus, the flattening of the slope is likely related to the burst of submesoscale. However, classical submesoscale theories, such as the surface quasi-geostrophic dynamics(Blumen, 1978; Held et al., 1995; Lapeyre, 2017) and the surface quasi-geostrophic dynamics with

ageostrophic advection (Boyd, 1992; Callies and Ferrari, 2013), predict a -5/3 and -2 kinetic energy spectrum, respectively. The flattening degree of the EKE spectral slope in the 2-km simulation has not reached a slope close to -2, which means that the current ISOM cannot sufficiently describe submesoscale processes.

**Table 3.** Statistics (maximum values and time-averaged spatial coverage) of the surface Rossby number for three resolutions in the Agulhas (Fig.6) and Gaussian plateau (Fig.7) regions for snapshots of 60 model days.

|              | Maximum | <0.2   | 0.2~0.5 | 0.5~1.0 | >1.0    |
|--------------|---------|--------|---------|---------|---------|
| 8km_Agulhas  | 1.64    | 86.24% | 13.13%  | 0.60%   | 0.03%   |
| 4km_Agulhas  | 2.71    | 73.19% | 23.79%  | 2.79%   | 0.23%   |
| 2km_Agulhas  | 6.79    | 50.96% | 36.17%  | 11.03%  | 1.84%   |
| 8km_Gaussian | 1.06    | 92.86% | 7.05%   | 0.09%   | <0.01%  |
| 4km_Gaussian | 2.42    | 82.88% | 16.01%  | 1.06%   | 0.05%   |
| 2km_Gaussian | 5.93    | 60.92% | 31.29%  | 6.89%   | 0.90%   |

In addition, in our early design (Tables S1 and S2 in the supplementary material), ISOM did not use KPP scheme, which resulted in underdeveloped mixed layers and highly intensified surface flow. Interestingly, it obtains the surface EKE spectrum that exhibits distinct submesoscale features at a scale of 8-30 km with a -5/3 spectral slope (Fig. S5 in the supplementary material) consistent with the surface quasi-geostrophic theory. Incorporating the KPP scheme promotes vertical mixing in the upper layer and weakens the possibly unrealistic submesoscale intensification phenomenon in the surface layer. Therefore, if researchers hope to improve the simulation of submesoscale based on the 2-km resolution ISOM in the future, we suggest introducing effective submesoscale parameterization schemes, such as Fox-Kemper and Ferrari (2008). Otherwise, one needs to adopt a finer horizontal resolution of at least 1 km, which can also effectively enhance the model performance of the submesoscale, especially the symmetric instability (Wei et al., 2024). Nevertheless, we emphasize again that since the goal of this work is the oceanic mesoscale, the presented results are sufficient to demonstrate that the model can fully describe mesoscale processes and generate a type of MODNS dataset.

Similarly, we take 1024 km zonal segments at given locations and compute the surface enstrophy (or relative vorticity variance) spectrum $Ens(k)$ (Fig. 5g-i). It shows a spectral slope of -1 for all simulations on large scales, and the dissipation effect removes enstrophy on small scales. The result is qualitatively consistent with Chassignet and Xu (2017) (their Fig. 23) who studied the high-resolution model on the Gulf Stream. The scale range with a -1 spectral slope expands with finer horizontal resolution. In the 2-km simulation (Fig. 5i), the range of -1 spectral slope can extend to nearly 10 km, which fully covers the deformation radius scale. Further examination of the potential enstrophy conversion term between the large-scale flow and eddies in the enstrophy budget shows a holistically forward potential enstrophy cascade (Figure S9). The result further confirms that the 2-km-resolution ISOM can generate a type of MODNS dataset.

**Table 4.** Same as Table 3 but for the normalized strain rate.

|              | Maximum | <0.2   | 0.2~0.5 | 0.5~1.0 | >1.0  |
|--------------|---------|--------|---------|---------|-------|
| 8km_Agulhas  | 1.85    | 86.85% | 12.52%  | 0.61%   | 0.02% |
| 4km_Agulhas  | 3.00    | 72.65% | 24.27%  | 2.84%   | 0.24% |
| 2km_Agulhas  | 7.55    | 44.91% | 42.56%  | 10.70%  | 1.83% |
| 8km_Gaussian | 0.99    | 93.66% | 6.25%   | 0.09%   | 0     |
| 4km_Gaussian | 2.66    | 83.36% | 15.52%  | 1.06%   | 0.06% |
| 2km_Gaussian | 6.05    | 57.78% | 34.73%  | 6.52%   | 0.97% |

### 3.3 Examples of multiscale eddy interaction

In this section, we examine the performance of the 2-km ISOM in simulating the evolution of eddying processes. We choose the Agulhas and Gaussian plateau regions to show snapshots of the sea surface variables. We especially discuss the interaction between the stationary eddy and the meandering jet in the Gaussian plateau region.

Figures 6 and 7 show sea surface snapshots of the 2-km ISOM in the Agulhas and Gaussian plateau regions, respectively. The variables include sea surface temperature (SST), free sea surface height (SSH), kinetic energy, the normalized relative vorticity, the magnitude of horizontal temperature gradient, and the normalized strain rate. The normalized relative vorticity ($\zeta/f$) is the vertical component of relative vorticity ($\zeta = v_x - u_y$) divided by the local Coriolis parameter ($f = f_0 + \beta y$). The Rossby number (Ro) is often defined as the absolute normalized relative vorticity ($|\zeta/f|$) and $Ro \sim O(1)$ refers to active submesoscale processes (Thomas et al., 2008; Schubert et al., 2020). Since both forms can describe the richness of submesoscale activities and the version without the absolute value can also reflect the sign of vorticity, we call the normalized relative vorticity as Rossby number in the manuscript for convenience. The normalized strain rate is defined as $\sqrt{(u_x - v_y)^2 + (u_y + v_x)^2}/|f|$.

Tables 3 and 4 present the statistics of Rossby number and normalized strain rate under three horizontal resolutions in the Agulhas and Gaussian plateau region. As the horizontal resolution improves, both variables undergo remarkable enhancement. Compared with the 8-km simulation, the maximum values of Ro and normalized strain rate in the 2-km simulation increased by 4.1 times in the Agulhas region and by 5.6 times and 6.1 times in the Gaussian region. In addition, the time-averaged spatial coverage of large Ro or strain rate (> 0.5) increases by tens of times from 8-km to 2-km simulation. The spatial coverage of large values in the Gaussian plateau region is lower than that in the Agulhas region at all resolutions (because of the higher latitude with a larger $|f|$ in the denominator) but has more prominent growth with resolution. These results indicate that although the current ISOM cannot sufficiently resolve submesoscale according to the spectral analysis in Section 3.2, there is still a significant improvement trend with finer resolution.

The following sections will further demonstrate the powerful capability of the 2-km-resolution ISOM 1.0 to express mesoscale-related processes and thus become a qualified mesoscale ocean DNS dataset.

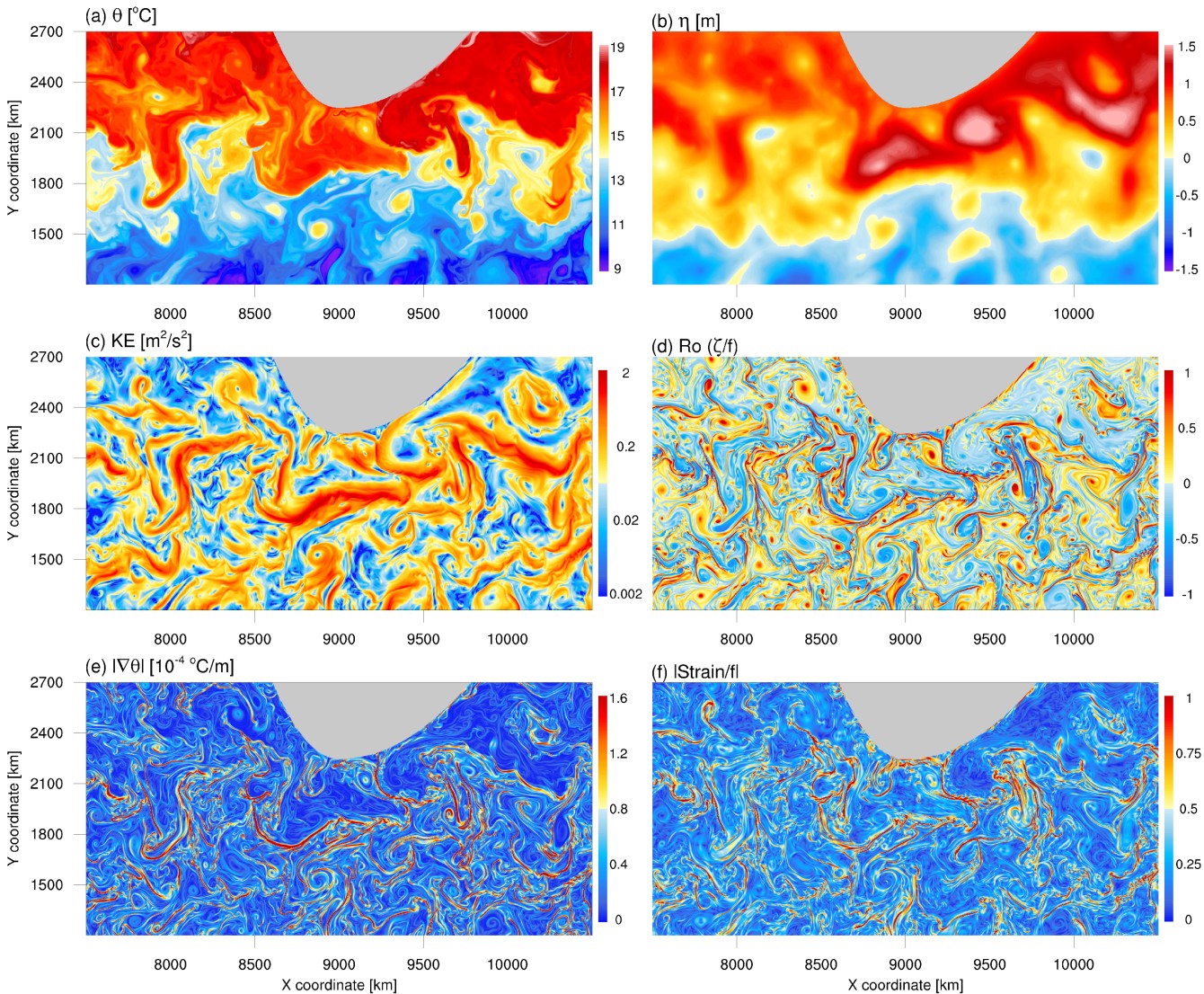

**Figure 6.** Surface snapshot of day 23 in the 8th model year of the 2-km simulation in the Agulhas region. (a) Potential temperature, (b) sea surface height, (c) kinetic energy, (d) Rossby number (normalized relative vorticity), (e) the magnitude of the horizontal temperature gradient, and (f) normalized strain rate.

### 3.3.1 Agulhas region

Though the Agulhas region (Fig. 6) is experiencing complicated processes, including mesoscale eddies, meandering jets, sub-mesoscale processes, and their multiscale interactions, our model successfully simulates the Agulhas retroflection (linked to the topographic steered large mesoscale eddies) and the eddy-shedding process of generating Agulhas rings (e.g., at x = 8400

km, y = 2500 km). These characteristics are consistent with Schubert et al. (2020), who used a regional model with $1/60°$ hor-
izontal resolution. The warm and cold water masses have extensive meridional displacement and drastically communicate with
each other, accompanied by multiple mesoscale eddies and vigorous jets (Fig. 6a, c). The existence of the topography makes
the southward intrusion of warm water on the eastern side more prominent, which is consistent with the realistic situation. The
warm eddies bear large size, and their flow matches the SSH field (Fig. 6b, c), but with loose internal structure and moderate
Ro in the core (Fig. 6d). The cold eddies have small scales but compact cores that often reach $|Ro| \sim O(1)$. The temperature
field between mesoscale eddies is severely stretched and deformed (Fig. 6e, f) to generate multiple fronts and filaments (Gula
et al., 2014; McWilliams, 2016).

Interestingly, our simulation captures the interaction among two large warm eddies and a small cold eddy adjacent to the
land at x = 9100 km and y = 2100 km. These two warm eddies exhibit complete and independent cores in the SSH field (Fig.
6a, b) and can be regarded as typical geostrophic eddies. Albeit squeezed by the large eddies, the small cold eddy keeps a
compact internal structure with large Ro in the core (Fig. 6d) and locally enhanced flow (Fig. 6c). Strong narrow temperature
fronts exist surrounding the warm eddies (Fig. 6e), with a width of less than 40 km and the most vigorous surface kinetic
energy in the region (Fig. 6c). The edges of mesoscale eddies have intense strain rate (Fig. 6f) and large Rossby numbers (Fig.
6d). The spatial pattern shows the clue of submesoscale symmetric instability with alternating signs and comma-like structures.
However, a horizontal resolution of 2 km (and without any specialized submesoscale parameterization) does not seem sufficient
to clearly resolve the symmetric instability. Even with the $1/60°$ horizontal resolution adopted by Schubert et al. (2020), the
development of symmetric instabilities is far from complete, compared to Gula et al. (2014) and McWilliams (2016) using
regional models with hundred-meter-level horizontal resolutions. Recently, a set of global simulations (Wei et al., 2024) based
on LASG/IAP Climate system Ocean Model of version 3 (LICOM3) show that when the horizontal resolution reaches $0.01°$,
the model's capability is explosively enhanced to express the symmetric instability and the 'submesoscale soup' (McWilliams,
2016) between mesoscale entities. The submesoscale level of the 2-km simulation in the manuscript basically conforms to the
$0.02°$ LICOM3 and the $1/48°$ LLC4320 MITgcm simulation (Su et al., 2018).

### 3.3.2 Gaussian plateau region

Figure 7 shows sea surface snapshots of the 2-km ISOM in the Gaussian Plateau region. We can see the topographic imprint,
the meandering jet, and the large eddies downstream. The meandering jet is reinforced and steered towards the southeast by
the topographic slope and generates a highly chaotic region with energetic eddy activities and potentially elevated mixing.
The warm water masses are carried southward by jets and eddies, and the cold water is drastically transported towards the
plateau from the south (Fig. 7a), which is consistent with the realistic situation near the Kerguelen Plateau. The SSH field
(Fig. 7b) implies the existence of energetic geostrophic currents (Fig. 7c) but is too smooth to reflect the diversity of multiscale
eddy interactions that are vividly shown in the Ro field (Fig. 7d). As the jet shoots downstream, intensive strain emerges and
drives rich submesoscale filaments along the track. However, the spatial pattern of the strain rate and temperature gradient near
the topographically reinforced jet differs from that around large mesoscale eddies. Multiple hotspots and chains with locally
excited strain rates and temperature gradients interweave the holistic network-like pattern within the meandering jet (Fig. 7e,

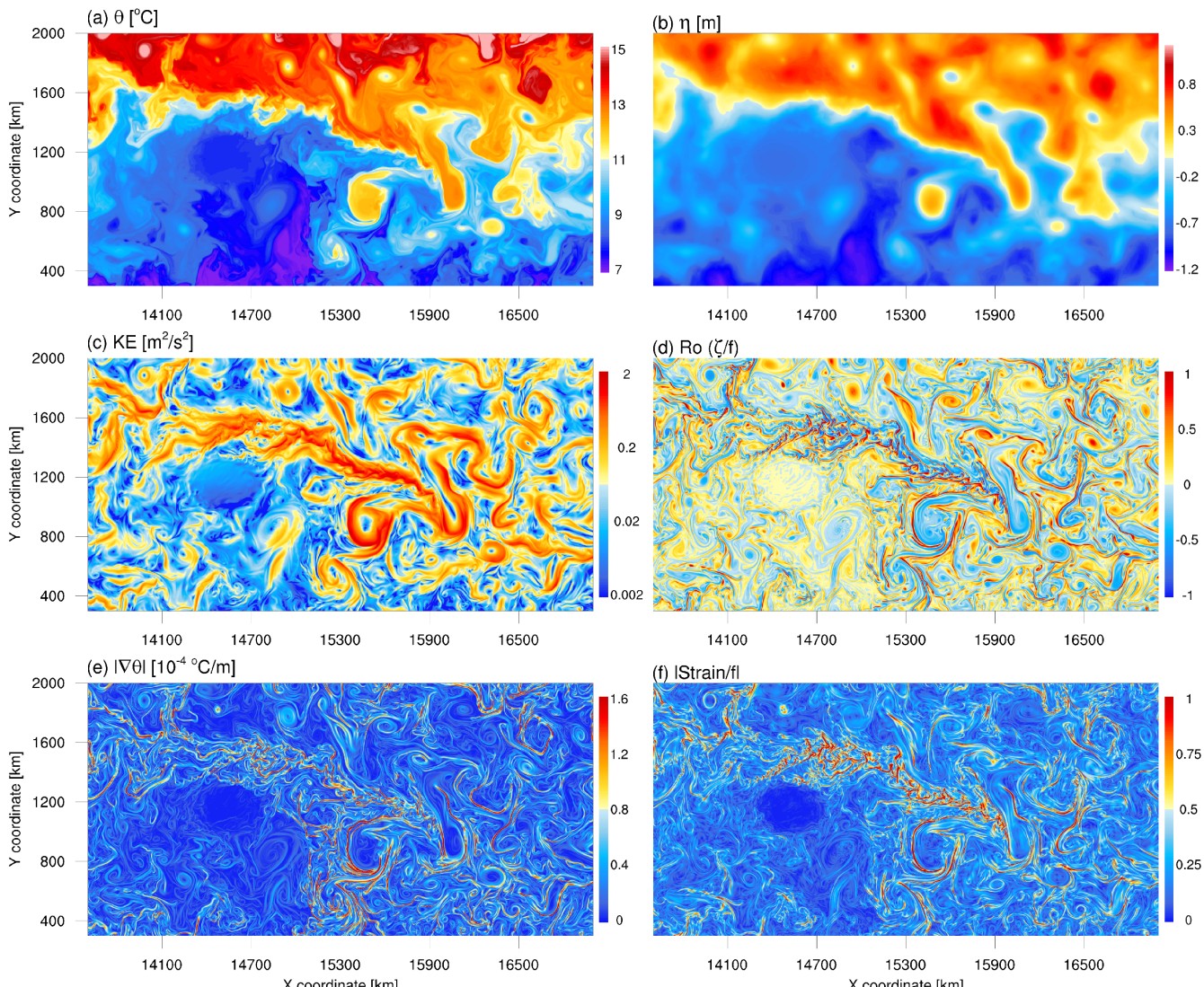

**Figure 7.** Same as Fig. 6 but for the Gaussian plateau region.

f). Rings or elongated lines distinctive from the surrounding environment exist around the mesoscale eddy in the Agulhas and Gaussian plateau regions.

Our model focuses on oceanic mesoscale processes (eddies and meandering jets). The mesoscale eddies can be categorized as transient eddies with shorter lifecycles and stationary eddies that are long-term and often related to topography (Bischoff and Thompson, 2014; Lu et al., 2016; Khani et al., 2019; Xie et al., 2023). Both of them substantially contribute to the eddy transport process. All types of mesoscale processes are shown vividly in Fig. 7, especially the stationary eddy downstream the plateau (at x = 15500 km, y = 900 km). We will zoom in on this area in Section 3.3.3 and demonstrate the evolution details

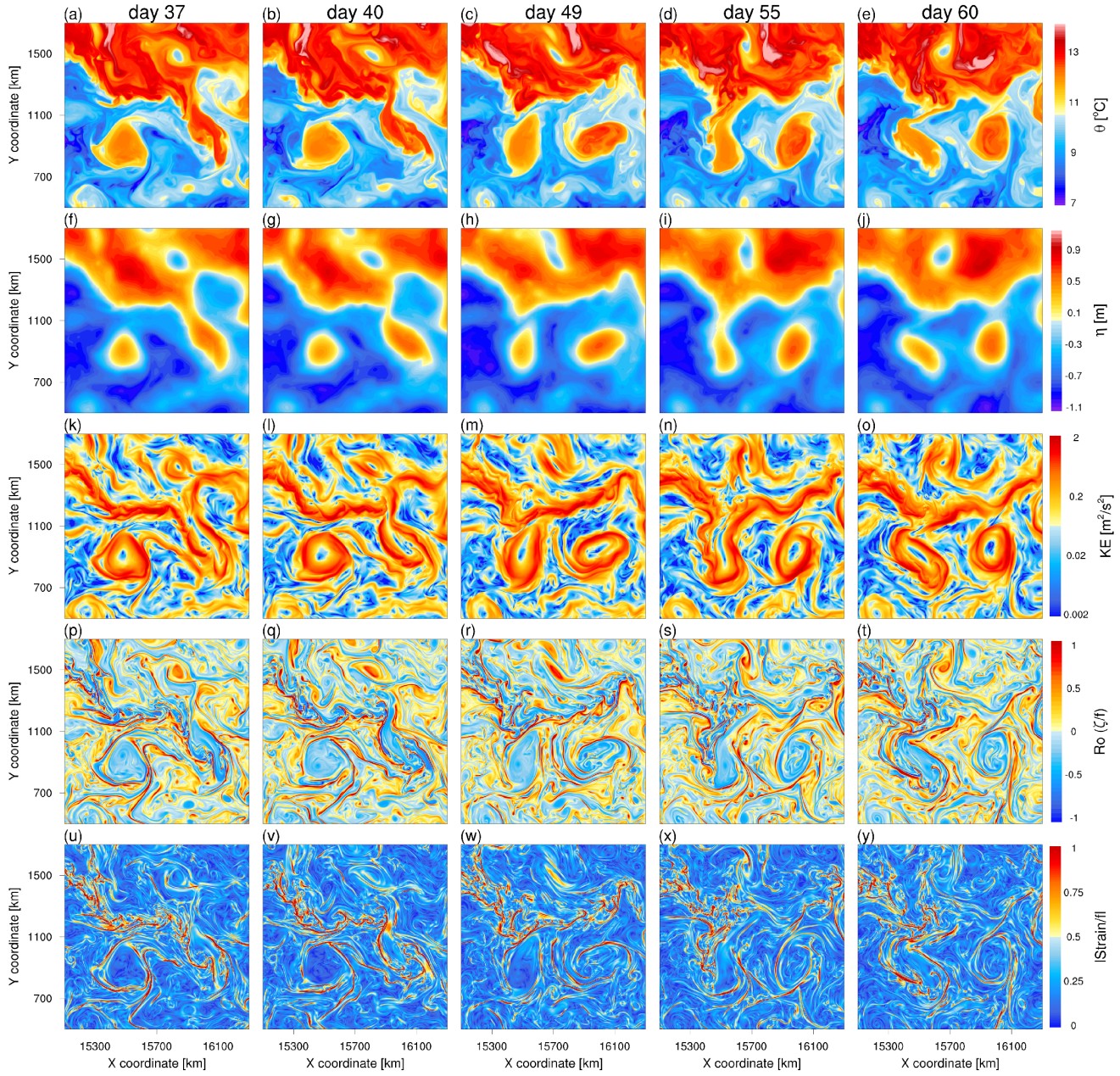

**Figure 8.** The evolution of stationary eddies, transient eddies, the meandering jet, and their interactions from the 2-km simulation in a zoomed region downstream of the Gaussian plateau.

of the flow to explore the complex multiscale interactions between stationary eddies, transient eddies, meandering jets, and submesoscale phenomena.

### 3.3.3 eddy-jet interaction

We now zoom in on a region east of the plateau in Fig. 7 and demonstrate the eddy-jet interaction using the model output in the 2-km simulation. In the period displayed, the jet is in a stage of drastic meridional movement, making it easy for eddy formation and intimate eddy-jet interactions.

On day 37, there exists an energetic jet along the temperature front with large meridional displacement, an individual warm mesoscale eddy with a diameter of over 350 km on the southern side, and a cold eddy on the northern side of the jet (Fig. 8a, f, k). From Rossby number and normalized strain rate (Fig. 8p, u), the jet had a network-like pattern mentioned in Section 3.3.2 upstream of $(x, y) = (15900$ km, 1000 km), where submesoscale signals are locally excited. However, downstream of this point, the pattern becomes a much simpler curved pattern similar to a mesoscale eddy, and another mesoscale eddy soon forms here later (Fig. 8f-j). Concerning the large eddy southwest to the jet, it has a prominent core on the SSH field (Fig. 8f) with vigorous circulation (Fig. 8k) that tightly locks in a bulk of warm water mass (Fig. 8a). It forms a ring with high strain rate and Ro in the periphery (Fig. 8p, u), but it is relatively hollow inside. Though this eddy may sometimes couple with the meandering jet during its lifespan, it can survive for over three months with almost unchanged geographical location (Fig. 8k-o and Fig. S1), making it a typical stationary eddy. The cold eddy north of the jet is a transient eddy prone to environmental flow due to its limited spatial scale and circulation intensity. It struggles to maintain its structure and move westward (Fig. 8p-t).

On day 40, the jet stream cannot maintain such a large meridional displacement and starts to shift to the zonal mode (Fig. 8l). A warm eddy begins to separate from the jet stream (Fig. 8b, g). The connection part becomes narrow and filamentary in the SST field and owns enhanced Ro and strain rate fields (Fig. 8q, v). The stationary eddy upstream is still intact and almost becomes a perfect ring.

On day 49, the jet stream finishes throwing out the eddy on the right and moves close to the stationary eddy on the left (Fig. 8c, h, m). They trigger a violent strain field and intense submesoscale signals in between but with undermined fields for the rest of the ring (Fig. 8r, w).

On day 55, an interesting phenomenon happens. The stationary eddy is temporarily open, and the meandering jet instantly generates a remarkable meridional displacement through the eddy circulation (Fig. 8n), shooting the network-like structure of Ro and strain rate field towards the west side of the eddy (Fig. 8s, x). However, the state is unstable and unsustainable because the submesoscale network can maintain the linkage between the upstream and downstream parts of the jet, thereby helping to stitch and rebuild the zonal geostrophic current and push the stationary eddy disconnect with the jet stream several days later (Fig. 8j, t, y).

Once the western stationary eddy forms and detaches from the jet stream, its geographical location barely changes for over three months (Fig. S1 in the supplementary material). It can maintain a complete eddy-like pattern most of the time. Even though the jet stream sometimes borrows the eddy circulation, the main body of the stationary eddy does not suffer a devastating consequence. The eastern warm eddy also generates from the excessive meridional variation of the jet and has a broader range of motion. Further downstream, as the jet stream bends and breaks, another warm eddy with the motion even more wide-ranging is also easy to form. These warm eddies and northern smaller cold eddies can form a spectacular asymmetric

eddy street with a submesoscale network on the upstream part of the jet and strain-related submesoscale activities around and between mesoscale eddies (e.g., snapshot shown in Fig. S2).

To sum up, the 2-km-resolution ISOM simulation demonstrates reliable capability in expressing mesoscale-related processes and thus is qualified to become a type of MODNS. We obtain qualitatively consistent features with the realistic situation and comprehensively illustrate the diversity of mesoscale processes and multiscale interactions. Previous works (e.g., Capó et al., 2021; Gula et al., 2022) reveal that the mesoscale-related strain is a crucial mechanism for submesoscale motions. We validate this point and intuitively uncover that submesoscale motions can bridge mesoscale entities and affect the eddy-jet interaction. Therefore, an appropriate representation of submesoscale effects on the mesoscale (i.e., MOLES schemes) in eddy-rich or eddy-permitting OGCMs (e.g., $0.1°$) is crucial for improving the simulations and cost-effective compared to using kilometer-scale or even finer horizontal resolution. ISOM 1.0 provides a platform for testing such parameterizations. The MODNS data based on the 2-km simulation can also serve as a benchmark for *a prior* and *a posterior* tests, thereby guiding modelers to employ suitable schemes.

## 4 Multipassive tracer tests

We use ISOM 1.0 to conduct multipassive tracer tests in this section. Multiple passive tracers can be leveraged as samples to construct an overdetermined linear system of equations to estimate the eddy transport tensor (Bachman et al., 2015, 2020). A two-dimensional diagnosis (i.e., on the neutral surface) requires at least two nonparallel samples, and a three-dimensional diagnosis (i.e., on the z-coordinate) needs at least three nonparallel samples (Xie et al., 2023). We conduct online tests of passive tracers using the 8-km simulation of the 52th to 55th model years (the flow is the same for all tracers). We uncover the properties of passive tracers in ISOM 1.0 and propose guidelines for selecting passive tracer combinations, thereby providing technical references for works that employ relevant methods for eddy transport diagnosis and parameterization design.

All passive tracers in the manuscript obey the advection equation without diffusion and source/sink terms (i.e., no restoration to a prescribed profile). That is,

$$\frac{\partial C_i}{\partial t} + \mathbf{u} \cdot \nabla C_i = 0 \tag{10}$$

The only difference among tracers lies in their initial fields. To ensure that the tracers are mutually independent (i.e., have very low spatial correlation), we recommend the initialization of a combination of four passive tracers as follows (Fig. 9):

$$C_1(x,y,z,t_0) = y/L_y + \sigma_1(x,y,z) \tag{11}$$

$$C_2(x,y,z,t_0) = sin(\pi y/L_y) + \sigma_2(x,y,z) \tag{12}$$

$$C_3(x,y,z,t_0) = sin(\pi x/L_x) + \sigma_3(x,y,z) \tag{13}$$

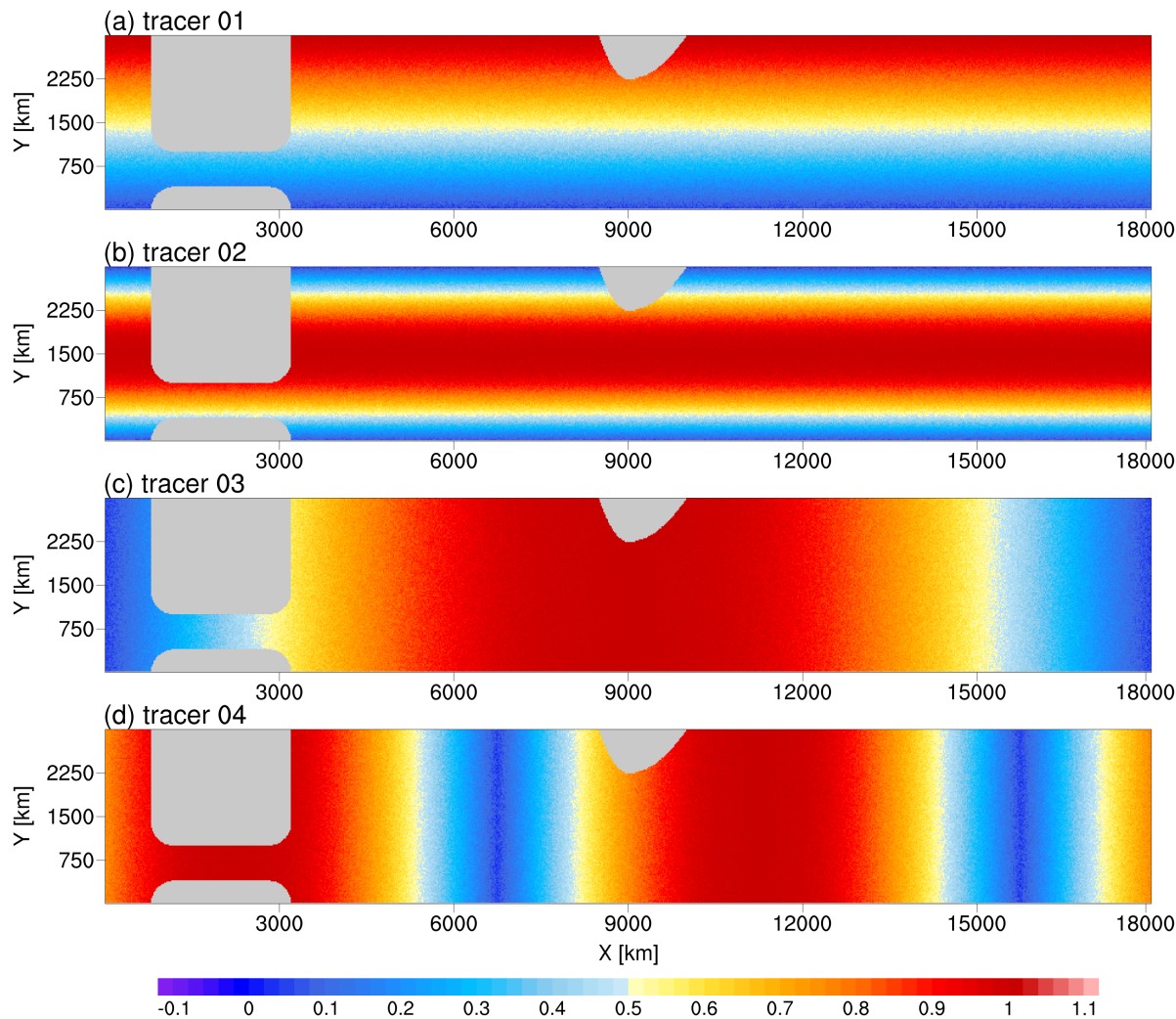

**Figure 9.** The initial fields for the recommended passive tracer combination.

$$C_4(x, y, z, t_0) = |sin(2\pi x/L_x + \pi/4)| + \sigma_4(x, y, z) \tag{14}$$

The random terms $\sigma_i$ $(i = 1, 2, 3, 4)$ follow a uniform distribution between 0 and 0.1. We further limit the initial fields within the range of 0 to 1. The setup ensures that the absolute Pearson correlation coefficients between tracers are far less than 0.1, and the initial fields can be considered mutually independent. Their spatial standard deviations are at the same level (approximately 0.3). This allows each tracer to contribute almost equally when solving the overdetermined linear system related to the transport tensor.

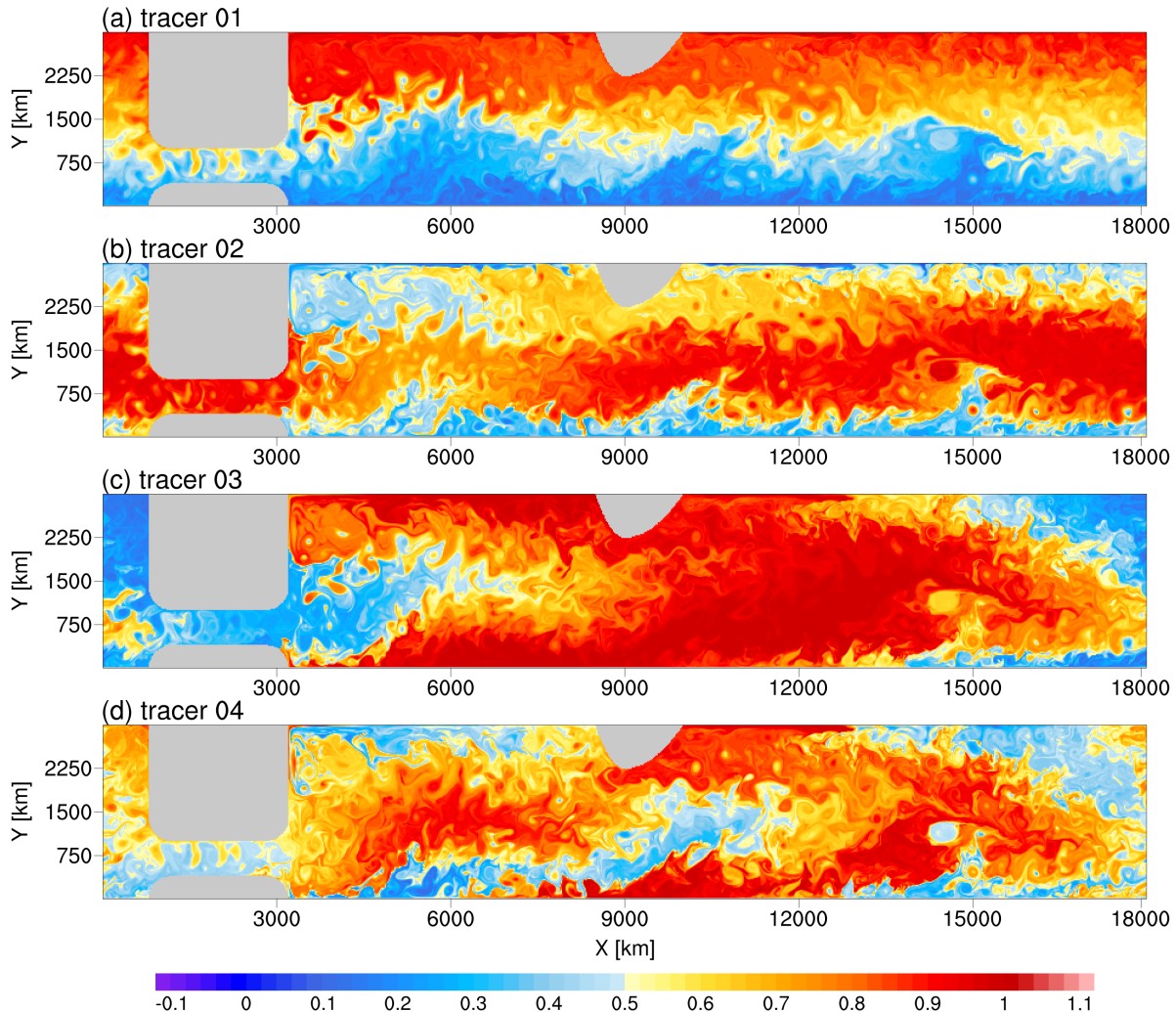

**Figure 10.** Snapshot of day 360 in the multipassive tracer experiment.

After long-term stirring by the flow, the homogenization of individual passive tracers occurs, and the spatial pattern among tracers becomes correlated (Figs. 10 and 11). This leads to the disappearance of the tracer gradient and the local alignment of the eddy flux vectors among tracers. Both effects can cause the failure of the multipassive tracer method for diagnosing transport tensors based on the flux-gradient relationship. There are two approaches to address these issues. The first approach is to add a restoration term to relax tracers to prescribed profiles. When diagnosing the transport tensor, one must address the problems caused by the restoration term (Bachman et al., 2015, 2020; Haigh and Berloff, 2021). The second approach is to continue to adopt passive stirring but release tracers several times(Wei and Wang, 2021). We prefer the latter approach and conduct validation tests. The key is to explore the time scale of homogenization and the correlation of the passive tracer

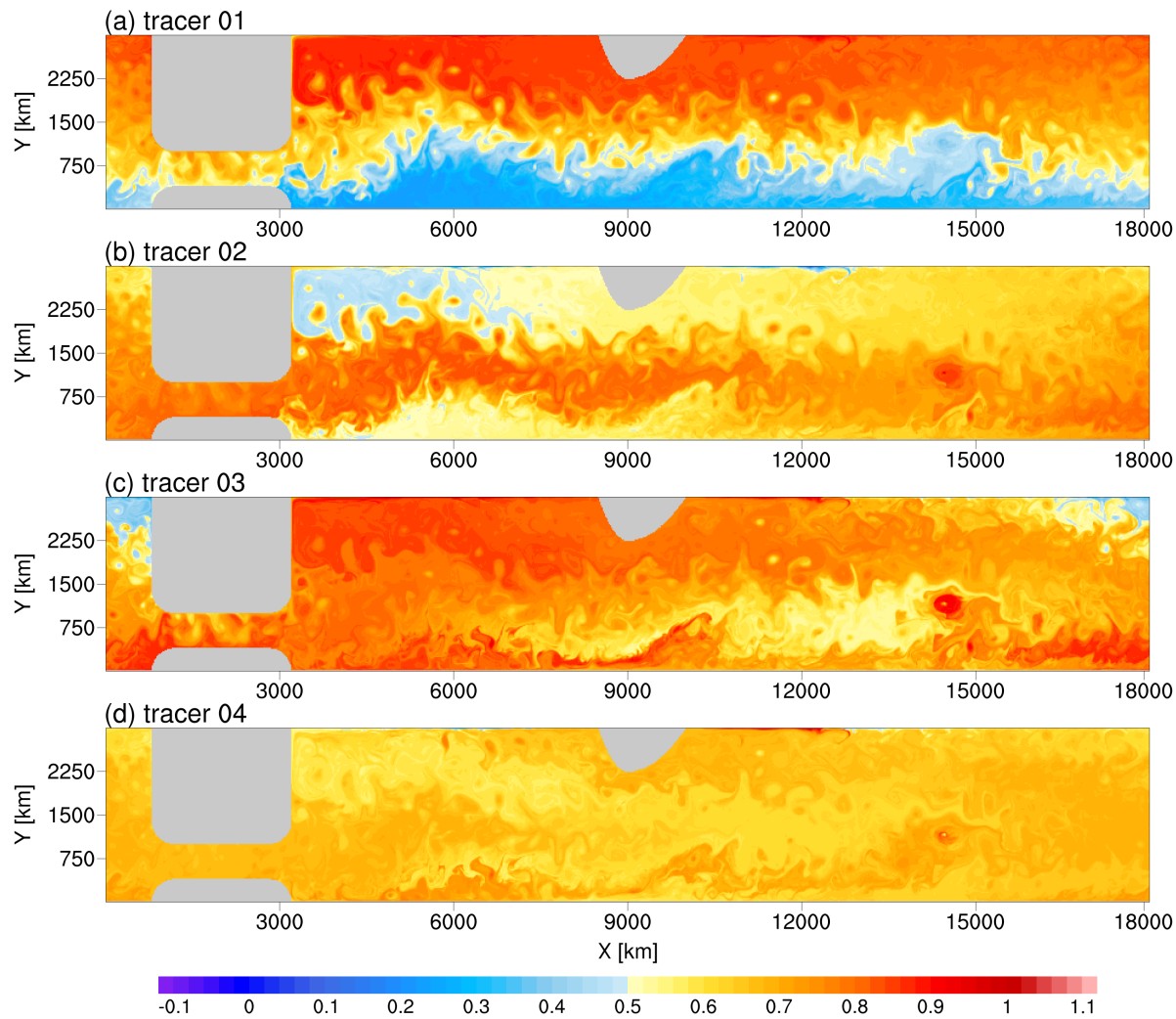

**Figure 11.** Snapshot of day 1440 in the multipassive tracer experiment.

combination. As long as the time interval of the release is shorter than the time scale at which unacceptable homogenization and correlation occur, the tracer output is suitable for subsequent diagnosis.

We test various initial tracer fields to explore how to delay the homogenization process and ensure the mutual independence of the tracers. We share noteworthy guidelines as follows.

(1) If the flow is quasi two-dimensional (e.g., large-scale oceanic motion), the initial tracer field should vary horizontally. One should avoid setting more than one initial field that changes only in the z-direction, such as $C(z) = |z|/H$ and $C(z) = sin(\pi|z|/H)$. Although these two expressions appear to be independent, they are highly correlated in the horizontal plane.

(2) If the domain has periodic boundaries, discontinuities in the initial fields at the corresponding boundaries should be avoided. For example, a passive tracer with an initial field of $C(x) = x/L_x$ seems not to cause significant numerical issues

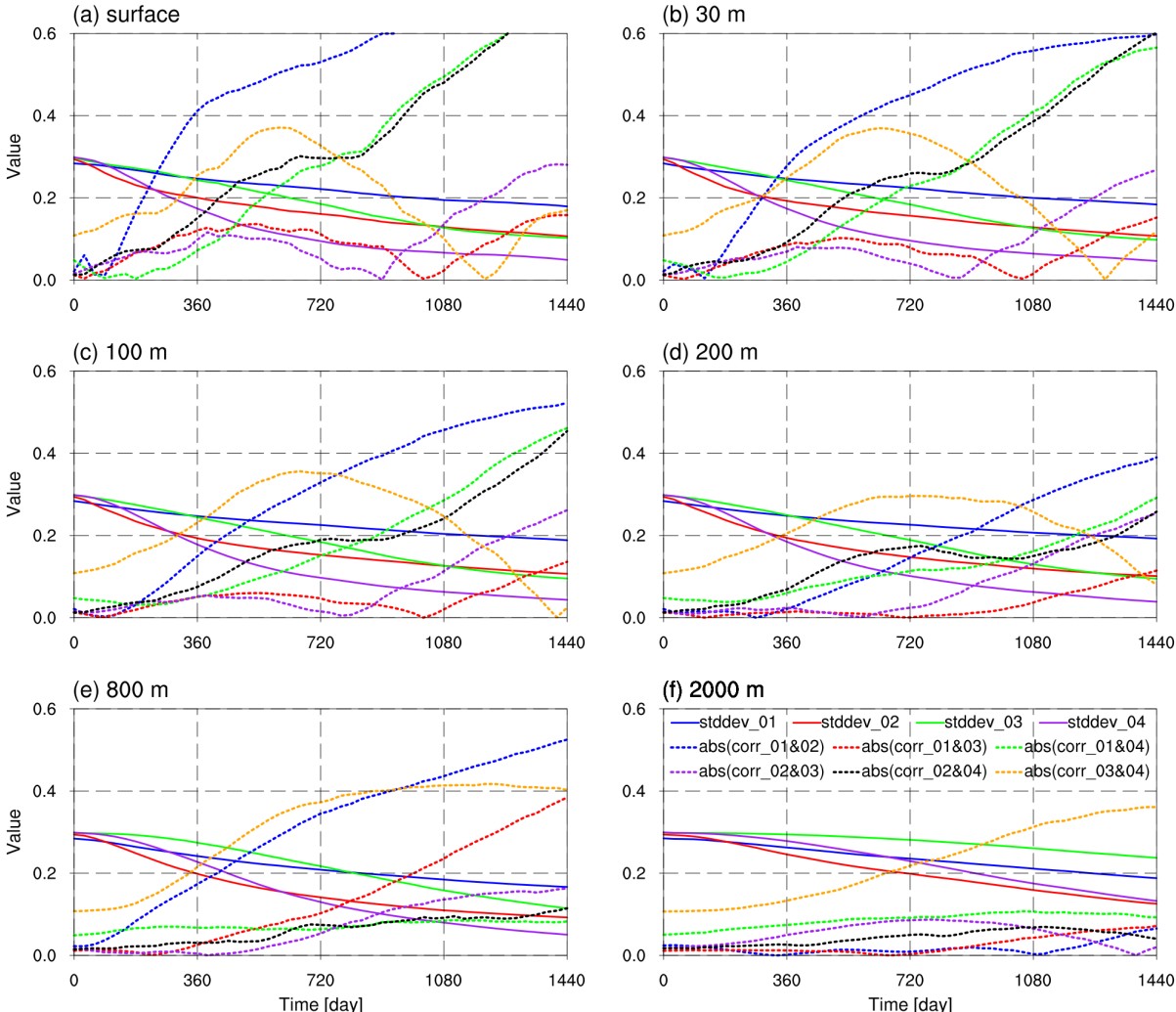

**Figure 12.** The temporal evolution of the standard deviation of tracers (solid line) and the absolute spatial correlation between tracers (dashed line) at different depths.

under high-order advection schemes, there is no guarantee that such an initial setting would not be problematic when using lower-order schemes or spectral methods.

(3) A crucial principle is that the initial tracer fields should be dominated by structures with large spatial scales. Otherwise, the tracer fields would be rapidly mixed by flow stirring. An extreme example is complete random initialization, which fills the tracer field with small-scale noise and can be homogenized within days. Another example is the initial field with two meridional half-period harmonic waves, $C(y) = |sin(2\pi y/L_y)|$. Though its spatial scale is far larger than that of random noise, it can be significantly mixed over a few months. Among the four tracers recommended in the paper, tracer $C_4$, which has an initial

field with two zonal half-period harmonic waves, homogenizes the fastest in the upper layer and reduces its spatial standard deviation by about 40% in the first model year (the purple solid line in Fig. 12). Tracer $C_2$, with an initial field of a meridional half-period wave, homogenizes the fastest in the deep and reduces its spatial standard deviation by about 50% within two model years (the red solid line in Fig. 12). The homogenization rates of tracers $C_1$ and $C_3$, which have larger spatial scales, are significantly lower (the blue and green solid lines in Fig. 12).

By further analyzing the temporal evolution of the absolute Pearson correlation between the four passive tracers (dashed lines in Fig. 12), we find: (1) the spatial correlation between tracers generally increases over time (though the evolution is not vigorously monotonic). This indicates that their spatial patterns tend to be similar under the continuous stirring of the same flow. (2) The evolution of correlations at different depths varies. Deeper locations generally have a slower increase in absolute correlation (though the monotonicity is not vigorously guaranteed). The increase rate in the upper mixed layer is much faster than in the deeper levels. (3) For convenience, we deem the absolute spatial correlation of less than 0.2 as low correlation. Within 360 model days, at least four tracer pairs for all levels and three tracers for levels under 200 m can maintain low correlation, and the spatial distribution of each tracer has not yet visually shown homogenization (Fig. 10). By the 720th model day, at least four tracer pairs can maintain low correlation for levels under the upper mixed layer. Therefore, under the condition of ISOM 1.0, controlling the duration of tracer release to less than 2 model years enables homogenization and correlation to be limited to an acceptable level so that the multiple tracer combination can serve as samples for accurately estimating the transport tensor.

## 5   Conclusions

In this paper, we have introduced an idealized Southern Ocean model (ISOM 1.0) that contains a simplified version of iconic topographic features in the Southern Ocean. We have conducted a fully eddy-resolving (2-km) simulation. The prominent feature of the model is the successful simulation of a fully developed and vigorous mesoscale eddying field. We reproduce the EKE spectrum of $k^{-3}$ predicted by geostrophic turbulence theory. In addition, the simulated ACC transport and geographical distribution of eddy activities are qualitatively consistent with the realistic situation. The model can also describe the topographic effect on stratification and large-scale flow and show reliable capability of simulating mesoscale processes.

To facilitate a smooth introduction of LES methods into ocean mesoscale parameterization, we propose the concept of MODNS. Its model grid should explicitly resolve the first baroclinic deformation radius, and the scales where dissipative schemes play a significant role are distant from the mesoscale dynamical regime, making it the benchmark for *a priori* and *a posteriori* tests of LES models or MOLES schemes into OGCMs. The 2-km idealized simulation satisfies the demands for MODNS and captures the submesoscale effects on the mesoscale entities. Therefore, it can serve as a type of MODNS and offer reliable data support for conducting relevant *a priori* and *a posteriori* tests of mesoscale-related parameterization schemes.

We demonstrate the diversity and high complexity of multiscale eddy interactions related to mesoscale processes by examining the evolution of mesoscale eddies, submesoscale phenomena, and meandering jets. We validate that mesoscale-related strain plays a crucial role in submesoscale processes. We also uncover that submesoscale motions can exert a bridge effect

between mesoscale entities and affect eddy-jet interactions. Therefore, we expect an appropriate expression of the collective submesoscale effects on mesoscale (e.g., MOLES schemes) in eddy-rich or eddy-permitting OGCMs to benefit the simulation of mesoscale variability.

We also use the idealized model to conduct multipassive tracer experiments. We reveal some guidelines for the initialization settings of passive tracers. We discover a combination of four passive tracers that can delay the homogenization process and ensure the mutual independence of tracers for a long time. With this combination and in ISOM 1.0, controlling the duration of each experiment of tracer release to less than 2 model years can ensure that the spatial standard deviation of the tracers and the correlations among the tracers are limited to acceptable levels. This allows the results from multiple release experiments of the

tracer combination to form a qualified sample for solving the eddy transport tensor based on the flux-gradient relationship. The tracer initialization guidelines can offer important references for similar ocean simulations.

The global ocean is a sophisticated system incorporating multiple spatiotemporal scales, and the large-scale oceanic dynamics vary across different regions. The idealized model in this paper provides a type of MODNS of the Southern Ocean with simplified topography and intermediate complexity. Similarly, one may design other idealized models, such as a double-gyre

basin or a cross-equatorial basin model with bottom topography and multiple vertical levels (e.g., Khani et al., 2019; Marques et al., 2022), to involve all mesoscale-related processes and generate corresponding MODNS datasets. The LES schemes and their matching parameters might differ if the background flow changes. Therefore, we should evaluate LES-related methods across various idealized and realistic models.

ISOM 1.0 uses the simple Laplacian and biharmonic dissipation schemes to handle the closure of the momentum equations.

Nevertheless, in the case of the kilometer-scale high resolution, employing a localized scheme (e.g., Leith, 1996) or a coupled anisotropic horizontal and vertical dissipation scheme (e.g., Khani and Waite, 2020) might potentially optimize the model performance near the grid scale range (e.g., submesoscale). In addition, the advection schemes can also influence the oceanic numerical simulations (Uchida et al., 2022; Thiry et al., 2024). Testing the impact of advection schemes on kilometer-scale high-resolution oceanic simulations is a challenging task that requires thorough planning and joint effort of the field.

Finally, we do not use any effective submesoscale-oriented parameterization in the simulation, and the model resolution is insufficient to explicitly resolve some crucial submesoscale processes (e.g., the symmetric instability) in the upper ocean. Thus, using MODNS data to design pure submesoscale parameterization schemes is inappropriate. Improving the horizontal resolution and optimizing the simulation of submesoscale processes in the mixed layer to generate the submesoscale ocean DNS is also an orientation for future work.

*Code and data availability.*   The dataset, the version of MITgcm package that we use to build ISOM, the relevant configuration files and codes needed for the simulations are publicly available in Science Data Bank (https://doi.org/10.57760/sciencedb.11634). The MITgcm software and documentation are available at http://mitgcm.org/, and the release is checkpoint67v-18-g1e25bc255. The Southern Ocean State Estimate is available online (http://sose.ucsd.edu/sose_stateestimation_data_05to10.html and http://sose.ucsd.edu/BSOSE6_iter106_solution.html).

*Author contributions.* JX was involved in conceptualization, conducting the simulation, programming, visualization, formal analysis, and writing the original draft. XW was involved in conceptualization, conducting the simulation, programming, and visualization. HL was involved in the conceptualization, formal analysis, supervision, and revision of the manuscript. PL was involved in conceptualization and formal analysis. JY was involved in conceptualization, formal analysis, and data processing. ZY, JW, and XH were involved in the implementation on HPCs. All authors contributed to writing and revising the paper.

*Competing interests.* The authors declare that they have no known competing financial interests or personal relationships that could have appeared to influence the work reported in this paper.

*Acknowledgements.* We express our gratitude to all the contributors to MITgcm and SOSE. We appreciate the patience and effort of the editor and reviewers. We thank Hao Fu for the enlightening discussion. We thank the support from the "Earth System Science Numerical Simulator Facility" (EarthLab) and the ORISE Supercomputer. This study was financially supported by the National Natural Science Foundation of China (92358302), the National Key R&D Program for Developing Basic Sciences (2022YFC3104802), the Tai Shan Scholar Program (Grant No. tstp20231237), and Laoshan Laboratory (LSKJ202300301).

*Financial support.* This study was supported by the National Natural Science Foundation of China (92358302), the National Key R&D Program for Developing Basic Sciences (2022YFC3104802), the Tai Shan Scholar Program (Grant No. tstp20231237), and Laoshan Laboratory (LSKJ202300301).

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
