# Peer review of "ISOM 1.0: A fully mesoscale-resolving idealized Southern Ocean model and the diversity of multiscale eddy interactions"

_Geoscientific Model Development, 2024_

## Author Response (AR1)

We thank the three reviewers and the editor for their comments and patience. Based on these helpful comments, we have significantly optimized the design of the idealized Southern Ocean model, resulting in simulation results that qualitatively align with the realistic situation. Since we need to redo the simulations, we take our time to conduct the new experiments and prepare the manuscript. Once again, we appreciate the editor and reviewers for their understanding.

Compared to the previous version, this work has undergone the following major changes:

(1) We have optimized the design of the idealized model. We expand the computational domain, deepen the bottom to 4000 m, and optimize the iconic topographic features to generate a flow field that better resembles the realistic situation, particularly by incorporating a deeper Drake Passage to enhance the ACC transport.

(2) We introduce the KPP scheme to implement a developed mixed layer.

(3) We provide sensitivity experiments on the flow response to wind stress profiles.

(4) We present a comparison of results with different horizontal resolutions and more detailed spectral analysis results.

(5) Since the experiments are redone, the case study in the previous manuscript has to be revised. In the new manuscript, we demonstrate multiscale eddy interactions by examining relevant variables in the Agulhas and Gaussian Plateau regions. Compared to the previous paper, we present more variables and focus on the eddy-jet interaction.

These revisions have made our model and the resulting data more reliable and also improved the readability and scientific content of the article. We hope the reviewers enjoy the reading process. Our previous response on the GMD discussion website was not based on the complete experiments and the finalized manuscript. Here, we will formally respond to the comments point by point in the following based on the existing results and the current manuscript.

===========================================================================

**RC1**

In this manuscript, the authors present an Idealized Southern Ocean Model ("ISOM 1.0") which is essentially an idealized configuration of the MITgcm. They argue that with a lateral resolution of 2km, this represents a type of DNS for the mesoscale ocean. The authors present some basic diagnostics for the mesoscale eddy energy and some typical eddy interactions. However, the focus appears to be primarily on the model configuration and the concept of ISOM as a tool, rather than original scientific results.

Overall, I am struggling with a recommendation for this manuscript. I have no fundamental issues with the overall concept, even though I disagree with some of the parameter choices - see below. The question is whether the concept is really that new? For example, the configuration, while differing in certain respects, is not so different to that of Neverworld.

In the conclusions, the authors state: "The prominent feature of the model is the successful simulation of a fully developed and vigorous mesoscale eddying field. We reproduce the EKE spectrum of $k^{-3}$ predicted by geostrophic turbulence theory. In addition, the simulated geographical distribution of eddy activities is qualitatively consistent with the realistic situation, and the model can describe the topographic effect on stratification and large-scale flow." However, I'm afraid that these results do not strike me as particularly novel.

On balance, I have suggested a major revision to give the authors the opportunity to respond. However, I would need to see original scientific results before I could recommend publication, or otherwise a really compelling case made for why the concept is fundamentally different to what has gone before and merits publication in its own right.

We thank the reviewer very much for the comments. The reviewer mentions that the focus of our manuscript is "*the model configuration and the concept of ISOM as a tool, rather than original scientific results*". Yes, this is our main objective, which is why we selected the Development and Technical Paper sector in GMD. Nevertheless, we have incorporated more original scientific results in the new manuscript, such as the sensitivity of the flow to wind stress profiles, the horizontal resolution effect, and the multiscale eddy interaction (e.g., the eddy-jet interaction).

In addition, the reviewer mentions that "*the configuration (of ISOM), while differing in certain respects, is not so different to that of Neverworld.*" To our knowledge, there are two versions of Neverworld. The early version of Neverworld (e.g., in Khani et al. (2019)), which contains simplified topography of the Southern Hemisphere. Neverworld-2 in Marques et al. (2022) is a cross-hemisphere domain that mimics the Atlantic Ocean. We believe ISOM has several distinctive aspects from the two.

(1) Though the topography in ISOM is idealized, it is more complex and realistic compared to the early Neverworld and Neverworld-2, focusing on the dynamics in the Southern Ocean.

(2) The early Neverworld uses a layer model (MOM) with only six constant density layers. While it is not entirely fair to directly compare the vertical grids of a layer model and a z-level model, having only six layers is insufficient to capture all the dynamics, particularly the mixed layer. This inadequacy is likely why Neverworld-2 employs 15 layers. The previous version of ISOM had 40 vertical levels. We now increase it to 75 levels with the usage of KPP scheme in the new version and improve the simulation of the mixed layer.

(3) The highest horizontal resolution of Neverworld in Khani et al. (2019) is 1/16 degree, which is sufficient to produce eddy-rich results for studying mesoscale eddy effects. However, it may not be adequate to explicitly resolve the deformation radius and fully capture the mesoscale-submesoscale interactions, and to obtain MODNS for the design of MOLES. Neverworld-2 in Marques et al. (2022) has the highest resolution of 1/32 degree, demonstrating the capability to capture the entire mesoscale dynamics. Nevertheless, Neverworld-2 oversimplifies the topography in the Southern Ocean and simulates an excessively strong ACC current. We have optimized the topographic settings in the new version of ISOM, significantly enhancing the simulation of ACC transport (increasing from 70 Sv in the previous manuscript to over 140 Sv).

Basically, we bridge the gap between model/simulation and "intermediate complexity" (quote from Marques et al. (2022)) for the Southern Ocean (Marques et al. (2022) seems to focus on the Atlantic Ocean). This is the merit of ISOM in terms of model configuration. As a tool, we aim for it to assist the scientific community in gaining a deeper understanding.

*Khani et al. (2019), Diagnosing Subgrid Mesoscale Eddy Fluxes With and Without Topography, JAMES*

*Marques et al. (2022), NeverWorld2: an idealized model hierarchy to investigate ocean mesoscale eddies across resolutions, GMD*

Specific comments:

Line 31: "as well as generating ..."

Line 35: A reference to Eden and Greatbatch (2008, doi: 10.1016/j.ocemod.2007.09.002) is also appropriate here.

Section 2.1: Please define all symbols used in the equations, including u, v, $\theta$, etc. You also switch from $\theta$ in equation 4 to T in equation 7, and it's unclear why this is named potential temperature rather than simply temperature given the linear equation of state.

Line 398: Space missing after "tensor".

Section 5: This should be written in the past tense, "In this paper, we have introduced ...", etc.

General comment: The paper is full of acronyms which do not help with its readability. Can the authors please work hard to reduce the number of these?

Thank you very much for the comments regarding the readability of the manuscript. We have addressed them.

Data availability: To be truly reproducible, the authors should provide the version number of MITgcm employed in this study.

We have provided a copy of the MITgcm package that we use, all the necessary configuration files, scripts, and pickup files on the website that distributes the datasets. Please refer to the code and data availability section.

Line 46: Can you explain what you mean by "its mathematical properties are not fully satisfied by the grid discretization of numerical models"? Does this imply something that is needed beyond the mathematical properties derived in Maddison and Marshall (2013, doi: 10.1017/jfm.2013.259), sections 2.3 and 3.3, and which hold for isopycnal thickness-weighted averaging?

Maddison and Marshall (2013) utilize and generalize the Eulerian-mean framework, but their framework is still Reynolds-averaged. The clue is that the example they provide is an ensemble average. The ensemble average is a classical Reynolds-averaged (RA) method that does not consider the situation of local scale separation. A good example is an ocean model with a horizontal resolution of (let us assume) 30 km. The model grid separates the entire process controlled by the primitive equations into a resolved part and a sub-grid scale part in a local way, and these two parts essentially interact with each other (cross-scale or multi-scale interactions). This is the scenario for the grid discretization of numerical models, especially when the grid scale is located in the eddying regime (not only for the ocean but also for models of other fluid dynamics). The RA-based as well as the Eulerian-mean framework are definitely self-consistent theoretical systems. If one uses them to diagnose something according to their interests, they can always find explanations for phenomena within the framework. However, the RA-based framework does not align with model practice. An extreme example is zonal average and time average (both are classical RA methods). If one uses the zonal average or time average to represent the resolved part of an ocean model, it is impossible to discern the behavior of the interactions between the resolved scale and sub-grid scale of the model. This does not mean RA-based methods are useless or incorrect; it just means RA-based methods are not suitable to guide sub-grid scale parameterization design and analyze interactive behavior near the model

grid-scale. I recommend Khani et al. (2019), Buzzicotti et al. (2023), and Xie et al. (2023). All works discuss relevant aspects in a local or coarse-grained sense (or personally, I prefer to call it in the LES sense), which is more aligned with model practice than RA-based methods.

*Khani et al. (2019), Diagnosing Subgrid Mesoscale Eddy Fluxes With and Without Topography, JAMES*

*Buzzicotti et al. (2023), Spatio-Temporal Coarse-Graining Decomposition of the Global Ocean Geostrophic Kinetic Energy, JAMES*

*Xie et al. (2023), A Multifaceted Isoneutral Eddy Transport Diagnostic Framework and Its Application in the Southern Ocean, JAMES*

Section 2.2: Why is the reference ocean depth so shallow at 3km, and why does the model Drake Passage shallows to 1km (by eye, the model Drake Passage looks even shallower in Figure 1a, but this might be an illusion?)

The authors subsequently find that the ACC transport is relatively weak at 65Sv, which I don't find surprising given the overly shallow reference ocean depth and Drake Passage depth (despite the authors' comment to the contrary on line 178). A realistic ACC transport should be a prerequisite and I suggest that the authors considering deepening their ocean to at least 4km, and the model Drake Passage to around 3km since there is circumpolar connection at the latter depth.

It is also worth noting that with the surface temperature relaxed strongly to 0 degrees at the south, and the northern vertical temperature profile prescribed through the sponge layer, the thermal wind is effectively imposed in the model:   what transport does this give, assuming no flow at 3km?

We thank the reviewer for this comment. In the previous manuscript, the reference ocean depth was set at 3 km. This choice was influenced by the use of several idealized channel models focusing on Southern Ocean dynamics that employed the same depth. We shallowed the model depth of the Drake Passage to 1 km based on the shallowness of the realistic Drake Passage outlet. However, as you and another reviewer pointed out, this depth was too shallow and inhibits the development of ACC transport. After testing, we finally set the reference depth to 4 km and adjust the highest bottom topography in the model Drake Passage region from 1 km to 2.5 km in the updated version of ISOM. This adjustment resulted in a time-averaged ACC transport of over 140 Sv, which falls within a reasonable range. Regarding the sponge layer, it is a mature practice in idealized Southern Ocean modeling works, e.g., Abernathey et al. (2011), Abernathey et al. (2013) and Bischoff&Thompson (2014). For ISOM, the setting of the sponge layer is to follow the realistic temperature profile in the north and, together with the surface relaxation boundary condition, to generate slope isopycnal/isothermal surfaces. The relation of zonal velocity and the stratification has been shown in Figures 2 and 4 in the new manuscript. For more information on the sponge layer technique, I highly recommend Abernathey et al. (2011) and Abernathey et al. (2013). In addition, the ACC transport not only depends on the prescribed stratification profile but is also highly sensitive to the surface wind stress. We have added content in Section 2.3 (Sensitivity of wind stress profile) in the new manuscript.

*Abernathey et al. (2011), The Dependence of Southern Ocean Meridional Overturning on Wind Stress, JPO*

*Abernathey et al. (2013), Diagnostics of isopycnal mixing in a circumpolar channel, Ocean Modelling*

*Bischoff&Thompson (2014), Configuration of a Southern Ocean Storm Track, JPO*

Line 147: Does "The topography within the passage has a piecewise linear depth" mean that the depth varies in a linear manner with partial cells, or that you are employing piecewise linear shaved cells? I think the former, but the text might be read as implying the latter.

We thank the reviewer for this comment. It is the former. We have revised the relevant content in lines 154-155 the in the new manuscripts: "The bottom topography within the passage rises from -4000 m to -2500 m within the range of x = 800 km to x = 2600 km and descends from -2500 m to -4000 m within the range of x = 2600 km to x = 3200 km."

General comment on Section 2 and Figure 1a: I'm not so sure the idealized topography is as novel as the authors imply. Many model studies have employed idealized Drake Paggase and/or ridge, e.g., the choices made here are not so different to those used in the two layer model of Tansley and Marshall (2001, doi: 10.1175/1520-0485(2001)031<3258:OTDOWD>2.0.CO;2).

As previously explained, our work bridges the gap between model/simulation and "intermediate complexity" for the Southern Ocean. This is the merit of ISOM in terms of model configuration. As for Tansley and Marshall (2001), we disagree with the comment. Our model is significantly more complicated than theirs, particularly in terms of topography and other model configurations. In addition, the new model can achieve simulation results that qualitatively align with the realistic situation.

Line 242: Why do you choose a 7th order advection scheme for the 8km simulation? This is indeed a very accurate scheme with minimal spurious diapycnal mixing, but I would not have expected that to be a major issue for such short integrations, unless you have a problem with undershoots, as described by Hecht (2010, doi:10.1016/j.ocemod.2010.07.005)?   I understand that the 3rd order DST advection scheme is chosen for computational efficiency at higher resolution - does this introduce any issues and, if not, then why not use it for the 8km integrations for consistency?

The reentrant channel case in MITgcm documentation uses the 7th-order scheme, and we followed it in the previous work for the 8km simulation. When the resolution becomes finer, we found it is too slow to integrate. Thus, we changed to DST-3 scheme. In the simulation of the new ISOM, we use DTS-3 scheme for all the simulations. Another reason to choose DST-3 scheme is because it is also very accurate, as you can see from Figures 2.14-2.16 in MITgcm documentation. As for Hecht (2010), the numerical effect discussed is mainly caused by the second-order centered-in-space advection scheme and is not that relevant to our work.

Line 297: The Rossby number is usually defined as a non-dimensional parameter quantifying the relative importance of the inertial and Coriolis accelerations, or of relative and planetary vorticity. Here, however, it is used as non-dimensionalized relative vorticity, which gives it a fundamentally

different meaning. Indeed, on line 300, the authors state that mesoscale processes dominates the flow when $|Ro| \ll O(1)$, but there are many regions (filaments) in Figure 6(d) where Ro=1 and yet submesoscale processes are prevalent. I therefore strongly advise avoiding this terminology and instead describing what is shown, i.e., $\zeta/f$.

The definition and usage of Rossby number in our work are reasonable and can be supported by numerous submesoscale-related studies from various academic journals. To clarify our definition and usage of the term, we have added the relevant content in Lines 372-377 in the manuscript: "The normalized relative vorticity ($\zeta/f$) is the vertical component of relative vorticity ($\zeta = v_x - u_y$) divided by the local Coriolis parameter ($f = f_0 + \beta y$). The Rossby number (Ro) is often defined as the absolute normalized relative vorticity ($|\zeta/f|$) and Ro ~ O(1) refers to active submesoscale processes (Thomas et al., 2008; Schubert et al., 2020). Since both forms can describe the richness of submesoscale activities and the version without the absolute value can also reflect the sign of vorticity, we call the normalized relative vorticity asRossby number in the manuscript for convenience."

We provide a list of several relevant works here:

*Su et al. (2018), Ocean submesoscales as a key component of the global heat budget, Nature Communications*

*Zhang et al. (2023), Submesoscale inverse energy cascade enhances Southern Ocean eddy heat transport, Nature Communications*

*Zhang et al. (2023), Ocean Modeling with Adaptive REsolution (OMARE; version 1.0) - refactoring the NEMO model (version 4.0.1) with the parallel computing framework of JASMIN-Part 1: Adaptive grid refinement in an idealized double-gyre case, GMD*

*Hohenegger et al. (2023), ICON-Sapphire: simulating the components of the Earth system and their interactions at kilometer and subkilometer scales, GMD*

*Schubert et al. (2019), Submesoscale Impacts on Mesoscale Agulhas Dynamics, JAMES*

*Naveira Garabato et al. (2022), Kinetic Energy Transfers between Mesoscale and Submesoscale Motions in the Open Ocean's Upper Layers, JPO*

*Schubert et al. (2020), The Submesoscale Kinetic Energy Cascade: Mesoscale Absorption of Submesoscale Mixed Layer Eddies and Frontal Downscale Fluxes, JPO*

*Luko et al. (2023), Topographically Generated Submesoscale Shear Instabilities Associated with Brazil Current Meanders, JPO*

*Balwada et al. (2018), Submesoscale Vertical Velocities Enhance Tracer Subduction in an Idealized Antarctic Circumpolar Current, GRL*

==============================================================================

**RC2**

Xie et al. develop a submesoscale permitting direct numerical simulation (DNS) of an idealized Southern Ocean. They show that by incorporating idealized topographic features and increasing the model horizontal resolution to 2km, they are able to represent some of the dynamic processes of eddy-eddy and eddy-jet interactions in the Southern Ocean. They argue that such DNS of oceanic mesocale processes will benefit the assessment of eddy parametrizations. While this manuscript does not present novel results in particular, I envision that the authors will use this as a stepping stone for future studies in developing and assessing

eddy parametrizations.

I would like to have the following points addressed by the authors before recommending the manuscript for publication:

We thank the reviewer very much for the patience and understanding of the purpose of our work. We have optimized the model configuration and redo the simulations. We achieve simulation results that qualitatively align with the realistic situation and offer more scientific results in the new manuscript, such as the sensitivity of the flow to wind stress profiles, the horizontal resolution effect, and the multiscale eddy interaction (e.g., the eddy-jet interaction).

Lines 88-89: The authors claim "We emphasize that the focus of the simulations should be on controlling the dynamics of the idealized model rather than on precise comparisons with observations or realistic model results." but without comparisons to direct observations, it would be difficult to assess the impact of advection schemes of momentum on the states realized by MODNS (e.g. Uchida et al., 2022, https://doi.org/10.5194/gmd-15-5829-2022); Thiry et al., 2024, https://doi.org/10.5194/gmd-17-1749-2024). Do the authors intend on testing different advection schemes as part of the ISOM database? It appears that even at 2km resolution, numerical simulations of the ocean are very sensitive to the advection schemes and/or closures used for surface convection.

We thank the reviewer for the comment and recommended papers. We agree that the issue is worth noting.

Even in the most idealized and simplest case, different advection schemes still have varying numerical effects. The collective effects generated by different sub-grid scale parameterizations and/or the same scheme under different parameter settings are often diverse. Therefore, it is understandable that different ocean models (with distinct advection schemes and parameterization settings) yield inconsistent simulation results. What makes the issue even more complicated is that interactions between the advection scheme and the sub-grid scale parameterization exist, such as the relationship between numerical mixing and physical mixing effects mentioned in Holmes et al. (2021). Though it is not directly related to this work, we are pleased to inform you that we are currently using a realistic global ocean model to evaluate the effects of advection and parameterization schemes.

Now back to MODNS. If we conduct systematic testing on the advection and closure schemes, it would be a huge project. Although I personally think this is meaningful, it goes far beyond the current scope of work. This means we must make some choices. Choosing means giving up something, which can sometimes make us and readers feel a bit discouraged, but that is just life. We tend to believe that between the advection scheme and sub-grid scale parameterization, the advection scheme has a more fundamental impact. Therefore, when generating MODNS at this stage, we prioritize choosing advection schemes that have excellent simulation performance in idealized examples such as 1-D and 2-D wave transport (such as DST-3 with a flux limiter, as you can see from Figures 2.14-2.16 in MITgcm documentation). As for the closure scheme, since the horizontal resolution itself is sufficient to ensure the natural development of mesoscale processes, the closure scheme of MODNS actually needs to deal with the collective effects of submesoscale and other fine-scale processes that the model cannot capture. The use of parameterization schemes that focus on different processes would obviously

lead to differences in the expression of these processes. But they are not something that MODNS needs to focus on (if one wants to better study these processes, the most important thing to do is to simulate them at higher resolution as some kind of DNS corresponding to physical processes). Therefore, we tend to use energy dissipation schemes with a clear effect that is easy to understand in MODNS.

In addition, regarding the comparison with observations, we agree with the reviewer's opinion. We may not have expressed ourselves clearly before. What we mean is that due to the idealization of the model, it is almost impossible to completely, quantitatively align with the observation. However, for such a model like ISOM, it is necessary to obtain reasonable results that are qualitatively consistent with the observations. We have clarified the relevant content in Lines 88-93 in the new manuscript: "Though it is fundamental for the simulation to conform to the realistic Southern Ocean in terms of basic dynamical features (e.g., quantitatively consistent with observed ACC transport value to provide a reasonable background flow for eddying processes), we emphasize that the focus of the simulation should be on controlling the dynamics of the idealized model rather than on precise comparisons with observations or realistic model results. We hope that the model can describe processes most closely associated with the mesoscale in the Southern Ocean, including mesoscale motions......" Based on this principle, we have reflected on our previous work after considering the opinions of all the reviewers. We think that the problem still lies in ensuring that ACC transport can be within a reasonable range. Therefore, we optimized the configuration of ISOM and obtained a time average ACC transport of over 140 Sv in the new simulation, which is within a reasonable range qualitatively consistent with the observation. As the subsequent questions are still related to this, we will provide relevant responses under the following comment.

*Holmes et al. (2021), The Geography of Numerical Mixing in a Suite of Global Ocean Models, JAMES*

Line 135: The vertical resolution is quite low for a model with the horizontal resolution of 2km (e.g. Balwada et al., 2018, https://doi.org/10.1029/2018GL079244; Ajayi et al., 2020, https://doi.org/10.1029/2019JC015827). Does this hinder the development of baroclinic instability and/or flow-bathymetry interactions in ISOM? Furthermore, the depth of 3000m is quite shallower that the actual Southern Ocean. Given this fact, does it make sense to compare the ACC transport to observational estimates as in Fig. 3?

We thank the reviewer very much for this comment. The setting of 3 km depth and 40 vertical levels is used in several idealized channel models focusing on Southern Ocean dynamics [e.g., Abernathey et al. (2011), Abernathey et al. (2013), and Bischoff&Thompson (2014)]. We followed them in the previous manuscript. When the reviewer pointed out the issue, we realized that we needed to step forward. We have increased the depth to 4 km and used 75 vertical levels (according to Stewart et al. (2017)) in the new ISOM. This helps to improve the simulation of ACC transport.

*Abernathey et al. (2011), The Dependence of Southern Ocean Meridional Overturning on Wind Stress, JPO*

*Abernathey et al. (2013), Diagnostics of isopycnal mixing in a circumpolar channel, Ocean Modelling*

*Bischoff&Thompson (2014), Configuration of a Southern Ocean Storm Track, JPO*
*Stewart et al. (2017), Vertical resolution of baroclinic modes in global ocean models, Ocean Modelling*

Line 184: Regarding the intense meridional displacement of the eddies and idealized ACC jet, have the authors played with different meridional wind profiles? In order to acheive a meridionally narrower ACC, some studies employ a quadratic form in sinusoid (e.g. Balwada et al., 2018, https://doi.org/10.1029/2018GL079244).

We thank the reviewer very much for the comment. We have used the new ISOM to test the influence of wind stress profiles. The ACC transport is sensitive to the wind stress forcing. When fixing the magnitude, the meridional wind profile with a quadratic form in sinusoid results in a narrower ACC stream but smaller ACC transport (135 Sv) compared to the normal one (145 Sv). This is because the total energy injected is less, especially in the model Drake Passage. Given the sensitivity, 2-D wind stress would be effective in fine-tuning the ACC transport and flow pattern. However, in such an idealized work like ISOM and Neverworld, it is not that necessary to involve such complexity. We have added Section 2.3 Sensitivity of wind stress profile in the manuscript to discuss the issue.

Section 3.2: The -3 slope is a prediction for the forward enstrophy cascade range but -5/3 is predicted for the inverse energy cascade range. Given that this is a modeling study, spectral cascades of enstrophy and energy should be also shown to convince the reader on the robustness of the spectral slopes.

We thank the reviewer for the comment. We have added the enstrophy spectrum in Section 3.2. We have also added the EKE and enstrophy spectrum results for simulations with different horizontal resolutions. The robustness of the results can be validated in the manuscript.

Section 4: Perhaps the authors have in mind to document the eddy transport tensor diagnosed from their passive tracers in another paper but I would like to see how the tensor changes in time as the homogenization of passive tracers takes place. If the tensor were to show strong time dependence, this would be an obstacle towards obtaining robust estimates of the tensor.

Putting the relevant work into another paper is indeed our plan. At present, we have not yet carried out work on the diagnosis of the transport tensor. However, this does not prevent us from discussing this issue here.

The main problem brought about by the homogenization of passive tracers is that the linear problem of solving the transport tensor becomes underdetermined. Therefore, more tracers are needed to figure out "how the tensor changes in time as the homogenization of passive tracers takes place". The combination of four passive tracers in this manuscript aims at maintaining overdetermination or at least well-posedness of the linear system during tracer release intervals (such as one year).

In addition, the time dependence of the transport tensor is inevitable because the eddying flow field varies in time, thus the transport efficiency must vary in time (e.g., Haigh et al. (2020)

and Haigh et al. (2021) show the snapshots of the transport tensor).

Regarding the robust estimates of the tensor, I guess what the reviewer might want to express is that different tracers may produce inconsistent estimation results for transport tensors (Uchida et al. (2023) and Xie et al. (2023)), namely, there may not be a universal/unique transport tensor for all variables. Personally, I think this is a pain point. I hope future works will provide an more explicit answer to this.

*Haigh et al. (2020), Tracer-based estimates of eddy-induced diffusivities, Deep Sea Research*

*Haigh et al. (2021), On eddy transport in the ocean. Part I: The diffusion tensor, Ocean Modelling*

*Uchida et al. (2023), Cautionary tales from the mesoscale eddy transport tensor, Ocean Modelling*

*Xie et al. (2023), A multifaceted isoneutral eddy transport diagnostic framework and its application in the Southern Ocean, JAMES*

===============================================================================

**RC3**

This work introduces an idealized Southern Ocean (SO) prototype at eddy-resolving resolutions (2 to 8 km) within the MITgcm model. The new model could successfully capture some mesoscale/sub-mesoscale features, including -3 upscale energy transfer, eddy-jet stream interactions and small/moderate Rossby number ranges by resolving Rossby deformation radius. However, some other SO features, such as ACC transport is underestimated in this new model. Also, a framework for passive tracer is provided.

This is a nice piece of work that provides dataset for high-resolution ocean modeling in the SO region. The manuscript is also well written and provided detailed information about their MODNS setup. To improve the usefulness of MODNS dataset, I think a few points need to be illustrated before I can recommend the publication of this work at Geoscientific Model Development. Here, I have itemized my major comments along with minor corrections:

We thank the reviewer very much for the comments and your patience. The underestimation of ACC transport is worth further optimization. Following all three reviewers' suggestions, we have optimized the model configuration and re-conducted all the simulations. We achieve simulation results that qualitatively align with the realistic situation and offer more scientific results in the new manuscript, such as the sensitivity of the flow to wind stress profiles, the horizontal resolution effect, and the multiscale eddy interaction (e.g., the eddy-jet interaction).

1. Figure 3: as it is explained in the manuscript the ACC transport is significantly underestimated in this setup. One possible reason for this performance is that the channel depth in the Drake Passage area is not realistic. I suggest the channel depth to be increased to be around 4 km and that the bottom topography can be set to be around 2.5 km height like the realistic Drake Passage region. I think this changes to the SO configuration setup would improve the poor estimate of ACC transport. I also suggest the author perform an investigation on changes of ACC transport with the surface wind stress and/or initial salinity profile.

We thank the reviewer for this helpful comment. We have optimized the model topography.

The reference domain depth has been set to 4 km as recommended. We have tested the setting of the model Drake Passage. We finally adjusted the highest bottom topography in the model Drake Passage region from 1 km (in the previous manuscript) to 2.5 km. The time-averaged ACC transport is now over 140 Sv, which falls within a reasonable range. Another reviewer has also suggested the depth of Drake Passage, and the specific number provided are deeper. Interestingly, we found that a situation between the two suggested values can give a reasonable result. We are very grateful for the suggestions from all the reviewers. We discuss the setting of the Drake Passage in Lines 159-171, please check. In addition, we will also add content about changes in ACC transport with the surface wind stress in Section 3.2 Sensitivity of wind stress profile in the new manuscript.

2. Lines 295-300: In this manuscript, Ro ~ 1 is introduced as an indication for submesoscale process. While moderate Ro might be a sign for smaller eddy-like features (in comparison with mesoscale range), but Ro~1 may not be considered as submesoscale "process". I suggest the authors find other metrics when they discuss submesoscale process (see e.g. McWilliams JC. 2016, Submesoscale currents in the ocean. Proc. R. Soc. A 472: 20160117, for more information).

We thank the reviewer for the suggestion and paper recommended. We have added more submesoscale-related indicators (e.g., the strain rate, the temperature gradient) and statistics in the revised manuscript. Please check Section 3.3 for details.

3. Equations (1) and (2): in actual LES/DNS setup for atmosphere and ocean, it is important to have a coupled (anisotropic) horizontal and vertical dissipation scheme in horizontal and vertical momentum equations (see e.g. Khani S, ML Waite 2020, An anisotropic subgrid-Scale parameterization for large-eddy simulations of stratified turbulence. Mon. Wea. Rev. 148, 4299-4311). I understand that if the authors might leave implementing this coupled/anisotropic setup for their future work, but I would like they add a short statement about this in their manuscript since their model setup is performed at very resolution 2 km.

We thank the reviewer very much for the understanding and the paper. We agree that a coupled (anisotropic) horizontal and vertical dissipation scheme can play a crucial role in high-resolution models. We have added the statement in the manuscript in Lines 576-579: "ISOM 1.0 uses the simple Laplacian and biharmonic dissipation schemes to handle the closure of the momentum equations. Nevertheless, in the case of the kilometer-scale high resolution, employing a localized scheme (e.g., Leith (1996)) or a coupled anisotropic horizontal and vertical dissipation scheme (e.g., Khani and Waite (2020)) might potentially optimize the model performance near the grid scale range (e.g., submesoscale)." Personally, I like Khani&Waite (2020). It would be very interesting to conduct an *a prior* test on the Khani&Waite (2020) scheme using MODNS data generated by ISOM. In addition, I also recommended the work of Khani&Waite (2020) to my collaborators who are planning high-resolution experiments using a realistic multi-resolution ocean model.

4. Figure 5: the horizontal axis in these plots is labelled as 'scale' which might be confusing. I

suggest using horizontal wavenumber k_h [1/km] (the authors might consider labeling the top horizontal axis with the 'horizontal scale [km]'.

We thank the reviewer for the suggestion. We have optimized Figure 5.

Minor comments:

Line 365: "submesoscale signals" ---> submesoscale filaments

Lines 397 and 400: put a space before '('.

I suggest a proof-reading before submitting the revision (there are a few grammatical typos in the text).

We thank the reviewer for the patience. We have improved the manuscript and enhanced the readability. If there still exist typos, please leave the comments.

---

## Author Response (AR2)

First and foremost, we express our gratitude to the reviewers and the editor for their patience and comments. We are delighted that the improvements we have made to the content of our study, particularly the model design and simulations, have been recognized. After the second round of reviews, we have only seen explicit comments left by Anonymous referee #2 on the GMD author interface. This document is our response.

**Referee #2**

This is my second time reviewing the manuscript. I appreciate the efforts taken by the authors in expanding their numerical simulation of an idealized Southern Ocean. The new set of simulations now align much closer to reality. On the point of passive tracers, I appreciate their reply regarding the eddy transport tensor. My previous point was more on how robust the tracer distributions remain orthogonal with each other when the end state of initial-value problems of passive tracers is universally a complete homogenization. Their Fig. 12 is informative in answering this question in demonstrating the time scales of passive tracer homogenization. I think the manuscript is close to publication and I only have a single point I would like to have addressed.

We thank the reviewer very much for the useful suggestions and patience, which helped us improve the research content and manuscript dramatically.

- Lines 359-365: This relation between KE and enstrophy spectra (Ens(k) = k^2 EKE(k)) is specific to two-dimensional (2D) turbulence. Given the first-order vertical isopycnal fluctuations in the Southern Ocean (e.g. Fig. 4), I find the justification to adopt spectral scaling laws of 2D turbulence difficult to justify. They should at least use quasi-geostrophic (QG) potential vorticity (rather than relative vorticity) if they wish to show the enstrophy spectra (Vallis, 2017; his Chapter 12). In my opinion, the authors should really just diagnose the QG enstrophy spectral flux and show whether the length scales they observe a -3 EKE spectral slope coincide with a forward enstrophy cascade.

The enstrophy (for the relative vorticity) spectrum has a spectral slope of -1 in the enstrophy inertial range, which is indeed the classical prediction of the 2D turbulence. We acknowledge that in the previous manuscript, we potentially assumed that the main processes simulated, especially the oceanic mesoscale processes, are quasi-2D, and therefore tend to approach the 2D turbulence. Another reason for choosing the current enstrophy spectrum is that we found that Chassignet and Xu (2017) provided the same (vorticity variance) spectrum for their simulation under similar horizontal resolution, and we did obtain qualitatively consistent results with theirs. After consideration, we decide to revise the content in the relevant paragraphs to avoid confusion between the concepts of geostrophic turbulence and 2D turbulence. The revised content in *Lines 359-366* in the new manuscript: "Similarly, we take 1024 km zonal segments at given locations and compute the surface enstrophy (or relative vorticity variance) spectrum Ens($k$) (Fig. \ref{fig:f05}g-i). It shows a spectral slope of -1 for all simulations on large scales, and the dissipation effect removes enstrophy on small scales. The result is qualitatively consistent with \citet{Chassignet2017} (their Fig. 23) who studied the high-resolution model on the Gulf Stream. The scale range with a -1 spectral slope expands with finer horizontal resolution. In the 2-km simulation (Fig. \ref{fig:f05}i), the range of -1 spectral slope can extend to nearly 10 km, which fully covers the deformation radius scale. Further examination of the potential enstrophy conversion term between the large-scale flow and eddies in the enstrophy budget shows a

holistically forward potential enstrophy cascade (Figure S9). The result further confirms that the 2-km-resolution ISOM can generate a type of MODNS dataset."

In addition, the reviewer hopes to verify the forward potential enstrophy (for potential vorticity) cascade in the simulation and has provided a relevant suggestion. The suggestion is thorough but not easily implementable at our current stage. Instead, we adopt a simple method, which is sufficient to illustrate that the potential enstrophy is transferred to small scales in our model. We diagnose the potential enstrophy conversion between the large-scale and eddy components of the flow and confirm that the simulation holistically exhibits a forward cascade of potential enstrophy. The relevant content is provided in Figure S9 in the supplementary material, and we also show it in the following.

**Figure S9.** The potential enstrophy conversion between the large-scale and eddy components of the flow, $-\overline{\mathbf{u}'q'} \cdot \nabla\bar{q}$ (unit: $10^{-30} m^2 s^{-3}$). $\mathbf{u}$ is the velocity. The potential vorticity $q = \frac{f+\zeta}{\rho}\frac{\partial\rho}{\partial z}$ as Holland et al. (1984). Bar and prime represent the Reynolds' time average and fluctuation, respectively. When the term is positive, there is a conversion of potential enstrophy from the large-scale to the eddying field on average (relevant discussion in Wilson and Williams (2004) or Eaves et al. (2024)). Though the term has spatial variation, the intense forward potential enstrophy transfer towards small scales can happen in regions with active eddy activities. The domain average of the term is also positive, which indicates holistically a forward potential enstrophy cascade in the model. [1] Holland, W., Keffer, T., and Rhines, P. (1984). Dynamics of the oceanic general circulation: the potential vorticity field. Nature, 308(5961), 698-705. [2] Wilson and Williams (2004). Why are eddy fluxes of potential vorticity difficult to parameterize? Journal of physical oceanography, 34(1), 142-155. [3] Eaves et al. (2024). An energy and enstrophy constrained parameterization of barotropic eddy potential vorticity fluxes. Authorea Preprints.

[Figure]

**Reference**

Chassignet, E. and Xu, X. (2017): Impact of horizontal resolution (1/12 to 1/50) on Gulf Stream separation, penetration, and variability, Journal of Physical Oceanography

Holland, W., Keffer, T., and Rhines, P. (1984). Dynamics of the oceanic general circulation: the potential vorticity field. Nature, 308(5961), 698-705.

Wilson and Williams (2004). Why are eddy fluxes of potential vorticity difficult to parameterize? Journal of physical oceanography, 34(1), 142-155.

Eaves et al. (2024). An energy and enstrophy constrained parameterization of barotropic eddy potential vorticity fluxes. Authorea Preprints.

---

## Author Response (AR3)

Thank you for resubmitting your manuscript and thoroughly addressing the reviewers' comments. Your revisions have improved the clarity and overall quality of the paper, making it a valuable contribution to our journal. I have reviewed the changes and appreciate the effort you have dedicated to enhancing your work.

The expanded methodology section is particularly well done, providing additional clarity regarding your experimental procedures and data analysis techniques. This significantly enhances the reproducibility of your study. However, there are still some areas where minor revisions could further strengthen your paper. Here are my suggestions:

First and foremost, we thank the topic editor very much for the patience and suggestions below. We are delighted that the improvements we have made to the content of our study, particularly the methodology section, have been recognized. We will reply to the comments point-by-point in the following.

Introduction:

While you have included several key references, incorporating more recent studies would provide a more comprehensive background and set the stage for your work. Consider discussing recent advancements and their relevance to your research to highlight its novelty.

We have added several studies published in 2024 to the Introduction part of the new manuscript.

Methods:

The added details are valuable; however, ensure that any specific software or equipment used is referenced with version numbers or specifications, as this will assist readers wanting to replicate your study.

We have added the relevant information in the code and data availability statement in Lines 591-592: "The MITgcm software and documentation are available at \url{http://mitgcm.org/}, *and the release is checkpoint67v-18-g1e25bc255*." Meanwhile, we also revised the reference of the MITgcm documentation to the same release checkpoint to ensure consistency.

Results:

Figures and tables are generally clear, though I suggest reviewing the legends for precision. For instance, Figure 3 would benefit from an expanded caption explaining the observed trends in greater detail.

Where applicable, ensure that all statistical analyses are fully described, with confidence intervals and p-values reported for key findings.

We have expanded the caption in Figure 3: "(a) The time series of ACC transport ...... The black line for the 8-km simulation starts from the 48th model year. *The ACC transport in the 8-km simulation has reached a statistically steady state in the 48th model year. After refining the horizontal resolution, the ACC transport can also maintain a statistically steady state close to the result of the 8-km simulation.*......"

Discussion & Conclusions:

The discussion is adequately expanded to address most of the reviewers' concerns; however,

integrating a comparison with other studies' findings, even those that differ from yours, will provide a balanced perspective.

Further, I recommend highlighting the broader implications of your findings and suggesting potential future research directions.

Currently concise, yet reinforcing the potential applications or real-world implications of your results would benefit the reader's understanding of your work's impact.

We have further revised the relevant part in the new manuscript.

References:

Ensure that all cited works in-text are fully accounted for in the reference list, and vice versa. Consistency in the citation format is crucial.

We have revised the reference of the MITgcm documentation to the same release checkpoint (checkpoint67v-18-g1e25bc255) to ensure consistency.